# "I KNOW THAT I DON'T KNOW... AND I EXPLAIN WHY" INTERPRETABLE ABSTENTION VIA COUNTERFACTUAL EXPLANATIONS

## ABSTRACT

Ensuring reliability in human-AI collaboration is crucial for fostering appropriate trust in hybrid decision-making systems, which hinges on performance and transparency, but also on understanding the limits of ML methods. Selective classification addresses this need by allowing classifiers to reject uncertain instances and focus on more confident predictions. However, very few works try to provide interpretable abstention policies for selective classification. In this work, we introduce a novel interpretable-by-design method for selective classification that leverages the distance between data points and their set of counterfactuals as a measure of uncertainty. By using this distance as a basis for rejection, our method formulates an effective abstention policy while providing contrastive and model-agnostic explanations. Experimental results indicate that our method effectively implements a rejection policy that is explainable by design without affecting performance.

## 1 INTRODUCTION

Over the past two decades, research in human-computer interaction has highlighted the importance of establishing an appropriate level of trust in artificial intelligence (AI) systems to ensure their safe and effective deployment in decision-making contexts (Lee and See, 2004), possibly enabling domain experts to take part in the design of human-AI hybrid systems. Performance and transparency are the two fundamental pillars on which we can build trust in AI systems: users look at accuracy metrics, confidence scores, and explanatory feedback to gauge whether an automated system's recommendations are worthy of acceptance (Zerilli et al., 2022). Communication of uncertainty status also plays a crucial role in enhancing transparency and enabling the integration of AI predictions with human judgment, while also facilitating humans to retain responsibility, agency, and control over the decision-making process (Bhatt et al., 2021). Explicit communication of uncertainty ratings or formulas like *"I know that I don't know"* helps align the perception of users with the actual system capabilities (Mehrotra et al., 2024). This is typically accomplished by implementing an abstention mechanism in the ML pipeline (Bhatt et al., 2021; Hendrickx et al., 2024): Learning to Abstain (L2A) systems equip a machine learning (ML) model with the ability to refrain from predicting when the uncertainty is too high or the cost of error is unacceptable (Punzi et al., 2025). By diverting ambiguous cases to human experts or stronger models, these systems reduce the risk of low-confidence mistakes. Despite their promise, most L2A methods adopt a black box rejection policy that does not disclose *why* certain cases were deferred. This opacity can erode trust in the system, as stakeholders are left without insight into the system's decisions and its rationale for abstaining (Artelt et al., 2023; Singla et al., 2023). Given that abstention mechanisms directly affect subsequent human workflows, the provision of interpretable rejections is essential: individuals require not only awareness of a model's uncertainty but also understanding of the underlying *reasons* for abstention, enabling them to determine whether to accept, contest, or act on the deferred case, without compromising trust in the AI system. To achieve this, human-centred approaches to AI show that explanations are judged by their contrastive, causal and actionable content rather than by low-level descriptions of model internals; explanations that answer "why this rather than that?" better align with how people seek reasons and build trust (Miller, 2019). Counterfactual, contrastive explanations have been advocated as a practical and actionable way to explain automated decisions and also to explain a model's confidence (Le et al., 2023).

In this paper, we address this topic by introducing an explainable by-design abstaining classifier named Selective Classification via Counterfactual Explanations (`SC-CE`, see Figure 1) Building on the selective classification paradigm, where the choice to reject is a function of input features and preliminary model outputs, `SC-CE` presents two key innovations: (i) it provides an interpretable quantification of model uncertainty through a confidence score based on the distance between each data point and its closest counterfactual, which is an approximation strategy to characterize the decion boundary of the ML model (Guidotti, 2024); (ii) it uses said counterfactuals to generate local explanations for the rejection of each case, offering users transparent insights into the inner workings of the abstention policy: in addition to acknowledging that it *knows to not know*, it also explain *why it is not confident enough*. Our rejection policy is therefore plug-in, model agnostic, and does not require access to any model information aside from its hard predictions.

The paper is organized as follows: Section 2 reviews related works on selective classification and explainable AI in the context of rejection policies. Section 3 presents the `SC-CE` methodology, detailing the problem setting, the rejection policy, and the process of generating explanations. Section 4 describes the experimental setup and results, followed by discussion and conclusions in Section 5.

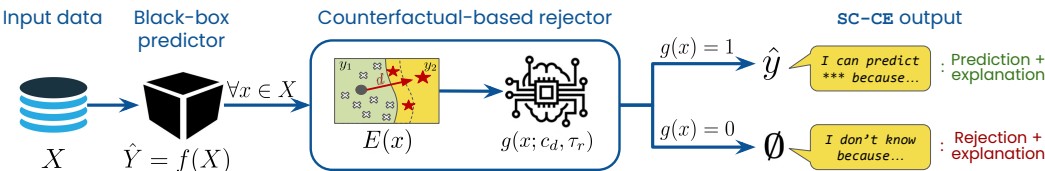

Figure 1: Graphical overview of `SC-CE`.

## 2 RELATED WORK

**Learning algorithms with a reject option**   The ability to incorporate in an algorithm the option to refrain from predicting on "difficult" instances, such as those with high uncertainty or where misclassifications carry critical consequences, has been extensively studied in the computer science literature. The first abstaining algorithms (Punzi et al., 2025) were introduced in the 1970s for classification tasks (Chow, 1970), giving rise to an entire research field known as *learning to reject* (L2R) (Zhang et al., 2023), *selective classification* (SC) (El-Yaniv and Wiener, 2010), or *machine learning with a reject option* (Hendrickx et al., 2024). While conceptually related and often interchangeably named, these approaches exhibit subtle algorithmic differences. L2R requires defining a class-wise cost function that penalizes both mispredictions and rejections (Cortes et al., 2023). In contrast, SC imposes different constraints: the rejection policy can be learned by either setting a target coverage and minimizing risk (bounded-abstention) (Geifman and El-Yaniv, 2019; Pugnana and Ruggieri, 2023a;b), or establishing a target risk and maximizing coverage (bounded-improvement) (Gangrade et al., 2020; Geifman and El-Yaniv, 2017). More generally, the goal of abstaining algorithms is to learn a model comprising two key components: a prediction function and a rejection policy (see Section 3 for the formal definition). These can be learned either independently or jointly, with varying levels of access to information depending on the specific architecture (Punzi et al., 2025; Hendrickx et al., 2024). The types of samples targeted for rejection typically fall into two main categories: ambiguities, where the model struggles to distinguish between multiple plausible classes, and novelties, which lie outside the distribution of known training data. A model with the reject option can also be formulated within the framework of conformal prediction (CP) (Linusson et al., 2018; Hallberg Szabadváry et al., 2025), which refers to a class of distribution-free statistical methodologies for assessing uncertainty in predictions. Unlike typical approaches that yield a single top prediction based on the highest confidence score, CP generates a set of predictions designed to include the true label with a user-defined probability level (Vovk et al., 2005). According to Linusson et al. (2018), confidence-credibility predictions may be used to establish a classification model with a reject option by assessing the cumulative error count on a set of predictions generated by a conformal classifier, ordered by their confidence levels. More recently, Hallberg Szabadváry et al. (2025) extends this formulation by integrating the distinction between ambiguities and novelties in the rejection policy.

**Explainable AI**   Explainable AI (XAI) encompasses a range of methodologies designed to enhance the transparency and interpretability of AI systems, particularly those relying on black-box machine learning models (Bodria et al., 2023; Guidotti et al., 2018). The primary goal of XAI is to support AI-driven decision-making by generating explanations that enable users to comprehend, trust, and assess model predictions. In critical domains such as healthcare and finance, explainability also plays a key role in promoting fairness, accountability, and adherence to regulatory standards (Ali et al., 2023). A common strategy in XAI involves deriving explanations from a surrogate model trained post-hoc to approximate the behavior of the original black-box system (Lundberg and Lee, 2017; Guidotti et al., 2024; Ribeiro et al., 2016). These post-hoc methods are generally categorized as either model-specific or model-agnostic, depending on whether they rely on the internal structure and training process of the black-box model. In particular, a model-agnostic approach provides explanations without accessing the internal workings of the model. Additionally, XAI techniques can be classified as either global or local: global methods aim to describe the overall decision-making process of the AI model, while local methods focus on explaining individual predictions (Bodria et al., 2023).

**Explainability of the reject option**   While L2A algorithms are designed to enhance user trust, they simultaneously introduce a new layer of opacity through their rejection policy, potentially compromising transparency and trustworthiness. Research addressing this limitation remains scarce. Most recently, Artelt et al. (2023) extends previous work (Artelt and Hammer, 2022; Artelt et al., 2022) to develop a local model-agnostic technique for post-hoc explanation after rejection: given a data point rejected by a conformal prediction rejection policy, this method generates a local surrogate model around that instance and uses this surrogate model to provide explanations (either counterfactual, semi-factual or factual). Our proposal extends this work by using the distance of counterfactuals from the original data points as the rejection criterion, thereby rendering our method model-agnostic and explainable by design. Related to our work are also the contributions by Singla et al. (2023), which refines an overconfident classifier using counterfactual explanations to improve uncertainty estimates, and by Lenders et al. (2024), who introduce a fair interpretable abstention classifier that provides some explainability for rejections; these explanations concern only rejections that are based on fairness concerns. In other words, the explanations for abstention provided by their method are rooted in local and global fairness analysis of the model's prediction and data.

## 3   SETTING THE STAGE

### 3.1   PROBLEM SETTING AND BACKGROUND

The goal of selective classification algorithms is to learn a model $f_g$ that consists of two components, namely a predictor $f$ and a rejection policy $g$. The former is defined as a function $f : \mathcal{X} \to \mathcal{Y}$, where $\mathcal{X}$ denotes the feature space and $\mathcal{Y}$ the target space, while the latter is generally defined (at inference time) as $g : \mathcal{X} \to \{0, 1\}$. In this paper we focus on dependent rejectors (Hendrickx et al., 2024) where the policy $g$ depends on the predictor $f$ through a confidence function $c_f : \mathcal{X} \to \mathbb{R}_+$ and a certain threshold $\tau_r$, such that $g = \mathbb{1}\{c_f(x) > \tau_r\}$ where $\mathbb{1}$ is the indicator function. The confidence function should provide a proper estimation for the true confidence of the model $f$ (Franc et al., 2023). The composed system is then defined by a function $f_g : \mathcal{X} \to \mathcal{Y} \bigcup \{\emptyset\}$ such that:

$$f_g(x) = \begin{cases} f(x) & \text{if } g(x) = 1, \\ \emptyset & \text{if } g(x) = 0. \end{cases} \tag{1}$$

Given an instance $x$, if the rejection policy $g$ rejects it, then no prediction is made; conversely, if $g$ accepts $x$, then the prediction function $f$ is applied to $x$ and the outcome $f(x)$ is observed. In this study, we assume $f$ to be a binary classifier. Ideally, $g$ should be able to accurately capture the decision boundary of $f$, rejecting the examples on which $f$ is prone to make mistakes while accepting those where a correct prediction is more probable. The policy $g$ can be learned with the bounded-abstention methodology (Pugnana et al., 2024; Geifman and El-Yaniv, 2017): let $\phi(g) = \mathbb{E}[g(X)]$ be a *coverage* function that computes the expected number of accepted instances of $g$ over some set $X$. Let $X_{train}$, $X_{cal}$, and $X_{test}$ be the training, calibration, and test sets sampled from some dataset $D$, respectively. We learn $g$ by sorting $X_{cal}$ with the confidence function $c$:

$c_{(1)} \leq c_{(2)} \leq \ldots \leq c_{(|X_{cal}|)}$; then select threshold $\tau_r$ corresponding to the percentile of a target coverage $\phi(g) = C$:

$$\tau_r = \inf\{\tau : \frac{1}{|X_{cal}|} \sum_{i=1}^{|X_{cal}|} \mathbb{1}\{c_{(i)} < \tau\} > (1 - C)\} \tag{2}$$

To keep this work self-contained, we present the formal definition of counterfactual explanation. Given an instance $x \in \mathcal{X}$, a classifier $f$, and a distance metric $d : \mathcal{X} \times \mathcal{X} \rightarrow \mathbb{R}_{\geq 0}$ (e.g., the Euclidean distance), a counterfactual instance is defined as an element of the set $X' : \{ x' \in \mathcal{X} \mid f(x') \neq f(x) \wedge x' = \arg\min d(x, x') \}$, and such that, for a given distance metric $d : \mathcal{X} \times \mathcal{X} \rightarrow \mathbb{R}_{\geq 0}$ (e.g. the Euclidean distance), $d(x, x')$ is minimal.

## 3.2 THE SC-CE METHODOLOGY

The objective of our methodology is twofold: (i) learn an interpretable rejection policy that determines whether test examples should be directed to the black-box classifier for prediction or not, and (ii) generate a human-interpretable explanation for the decision to abstain. We propose to implement a *dependent* rejection policy based on a confidence function that quantifies the uncertainty of the prediction of a classifier on a given instance by measuring its proximity to its closest counterfactual. Intuitively, finding a counterfactual near the decision boundary should be "easy", due to the relatively short distance that needs to be traversed to transition from one side of the decision boundary to the other. In turn, this short distance suggests a high level of uncertainty in the classifier, which can be interpreted as a reason for abstaining from making any prediction, i.e., if an instance and its counterfactual are too similar, the machine is likely unsure in the prediction. Conversely, if the distance of a test example from its counterfactual is sufficiently large, it can be inferred that the classifier possesses a considerable level of confidence in its prediction. Having this formulation, we are able to build a rejection policy without accessing the classifier's prediction probabilities. While prediction probabilities are usually available, they may not be so in contexts where a model is provided by third-party entities that keep it closed-source or when the model is protected for privacy reasons. Our methodology makes it possible to both explain and equip any model with a rejection policy to bolster its performance on the accepted instances. The complete SC-CE pipeline requires only the hard prediction queries $f(x) \in \mathcal{Y}$, without the need for gradient information, probability score, or architectural knowledge.

### 3.2.1 LEARNING THE REJECTION POLICY

The conceptual basis of our framework is that the distance between an input instance and its counterfactual explanations can serve as a proxy for the confidence score of a black-box model, hence enabling the computation of an accurate rejection policy. In SC-CE, the definition of the confidence function $c_f$ that determines the rejection policy $g$ depends on the generation of counterfactual instances $X'$. More precisely, given a counterfactual explainer (Guidotti, 2024):

$$E : \mathcal{X} \rightarrow \mathcal{X}, \qquad E(x) = X'_x = \{x'_1, \ldots, x'_h\},$$

SC-CE computes $d(x, x')$ for all pairs $(x, x')$, $x' \in X'_x$ and then derives a surrogate confidence $c_d$ for the model on $x$ by applying a user-selectable aggregation function $a$ (e.g. $\min, \max, \text{mean}$) over the set of distances:

$$c_d(x) = a\big(\{ d(x, x') : x' \in X'_x\}\big). \tag{3}$$

Note that whenever the counterfactual instances in $X'$ are chosen to minimize the distance function $d$ used to compute $c_d$, then they should all have the same distance from $x$; hence, Eq. equation 3 simplifies to $c_d(x) = d(x, x'_1)$. However, since different metrics can be employed to estimate minimality in the counterfactual extraction stage, we keep the more general formulation, allowing a flexible choice of aggregation. The same simplification applies when the explainer $E$ returns one single valid counterfactual. We then use $c_d(x)$ to inversely rank a calibration set, following Eq. 2:

$$\tilde{C}_g(\tau) = \frac{1}{|X_{\text{cal}}|} \sum_{x \in X_{\text{cal}}} \mathbb{1}\{c_d(x) \geq \tau\},$$
$$\tau_r = \inf\{\tau : \tilde{C}_g(\tau) \geq C\} \tag{4}$$

Note that the above can also be formulated in terms of the rejection (or abstention rate), which is simply defined as $r_g(\tau) := 1 - \tilde{C}_g(\tau)$. This modular design allows us to plug in *any* black-box counterfactual explainer $E$ (for instance, LORE, DiCE, etc.), *any* distance metric $d$ (such as Euclidean, Manhattan, mixed, or feature-weighted), and *any* aggregation policy $a$ (minimum, maximum, or mean).

Finally, the rejection policy $g$ is formulated so that data points whose distance from their closest counterfactual is less than the specified threshold $\tau_r$ are deemed too uncertain to be predicted by the classifier $f$ and are thus rejected:

$$g(x; c_d, \tau_r) = \begin{cases} 1 & \text{if } c_d(x) \geq \tau_r, \\ 0 & \text{otherwise.} \end{cases} \tag{5}$$

We note that, following Eq. 1, the policy returns 1 if an instance is accepted. Therefore, the distance computed in Eq. 3 must be greater than the threshold to accept an instance. If $X'_x = \{x'\} \implies |X'_x| = 1$, SC-CE can also be formulated in terms of conformal prediction by setting the non-conformity to $S(x) = \frac{1}{d(x, E(x))} = \frac{1}{d(x, x')}$ and then using the rejection rate as the $p$-value. We provide the exact percentile and $p$-value recipe in the Appendix A.

### 3.2.2 GENERATING EXPLANATIONS OF REJECTION OUTCOMES

The output of SC-CE provides the user with complementary pieces of information that, taken together, support an informed interpretation and appropriate use of the system: *i)* the classifier's predicted label $\hat{y}$, *ii)* the decision of the rejection policy $g(x) \in 0, 1$, and *iii)* a multi-modal explanation of the rejection policy's outcome.

The third component is the distinctive feature of SC-CE. Whereas conventional selective classifiers merely report whether the confidence in the most likely outcome lies above or below a threshold, our method additionally explains why the confidence measure falls on one side of the threshold for a given instance. Because the cutoff is calibrated according to the distance between an instance and its counterfactuals, these counterfactuals can be directly used to interpret the abstention policy. Specifically, for each rejected instance, SC-CE provides a multi-modal explanation that combines textual and visual elements: a brief message informs the user that whether the model is sufficiently confident to make a prediction, a bar plot shows the position of the instance-counterfactual distance relative to the rejection threshold, and an additional panel illustrates a minimal change to the input that would lead to an alternative prediction, thereby offering insight into the model's rationale for rejection.

Importantly, SC-CE does not use counterfactuals in the conventional sense, namely, to suggest minimal changes needed to make a rejected instance accepted. Because counterfactuals are constructed to lie close to the decision boundary, they often illustrate proximity to the boundary and therefore typically lie in the model's uncertainty region. However, in our formulation, this depends on the aggregation $a$, the chosen distance $d$, and the local shape of the decision boundary; thus, counterfactuals may but do not necessarily always trigger rejection themselves. We therefore present counterfactuals as visual/contrastive indicators of boundary proximity rather than as guaranteed rejected instances.

## 4 EXPERIMENTS

Through our experiments, we directly address the following research questions:

**RQ1** Given a data point, can its distance from an appropriate counterfactual be considered a good proxy for the confidence of the ML model?

**RQ2** Does SC-CE achieve comparable results to state-of-the-art selective classifiers in terms of predictive performance (i.e., non-rejected accuracy)?

**RQ3** Can SC-CE provide human interpretable explanations of the ML algorithm about the rationale behind the reason to abstain from making a prediction?

|              | *Accept* | *Reject* |
| ------------ | :------: | :------: |
| *Correct*      | CA | CR |
| *Misclassified* | MA | MR |

$$NA = \frac{CA}{CA + MA} \in [0, 1].$$

$$CQ = \frac{CA + MR}{CA + CR + MA + MR} \in [0, 1].$$

$$RQ = \left(\frac{MR}{CR}\right) \bigg/ \left(\frac{MR + MA}{CR + CA}\right) \in [0, +\infty].$$

Table 1: Left: Confusion matrix for a selective classifier. Right: formal definitions of the evaluation metrics used in the experiments.

## 4.1 EXPERIMENTAL SETTING

**Dataset and classification models**   We evaluate `SC-CE` with standard benchmark datasets on binary classification with synthetic and real data: `Adult` (Becker et al., 1996), `German` (Dua and Graff, 2019) , `Wisconsin` (Street et al., 1993), and `Two-Moons`.These datasets are openly available benchmarks commonly used in the evaluation of selective classification algorithms (Pugnana et al., 2024). Each dataset was partitioned into training, calibration, and test subsets, with the training subsets employed to fit four distinct classification models (`LightGBM`, `Multilayer Perceptron (MLP)`, `Random Forest`, and `XGBoost`) using 5-fold cross-validation for hyperparameter optimization. Detailed specifications of the hardware setup used for our experiments and a statistical description of the datasets used can be found in the Appendix C.

**Counterfactual generators**   We validate our framework with different methods for counterfactual generation: `DiCE` (Mothilal et al., 2020) balances proximity and diversity,`LoRE` (Guidotti et al., 2024) builds local decision tree surrogate models, and `ILS` (Piaggesi et al., 2024) which computes a latent space $L$ through a linear interpretable transformation $(\mathcal{X}, \mathcal{Y}) \to \mathcal{L}$ and then uses an encoder-decoder architecture to generate counterfactuals in $\mathcal{L}$. As the space $\mathcal{L}$ maintains semantic validity, a comparison between an instance and its counterfactuals can be computed both in the input and latent spaces. In our work, we refer to these variants as `ILS` and `ILS`$_{latent}$, respectively.

**Distance metrics**   To quantify the relationship between an instance $x$ and its counterfactuals $x' \in X'$, we evaluate several distance metrics. Indeed, each formulation captures different aspects of the relationship between instances and their counterfactuals. For example, $L_1$ and $L_2$ norms measure absolute differences, while cosine distance captures orientation differences regardless of magnitude. Given the pivotal function of this component in the `SC-CE` framework as proxies for model confidence, we assess several formulations of distances to verify our first reasearch question and determine which distance best correlates with the model confidence. After preliminary experiments, we selected a subset of metrics that exhibited the most promising results. $L_2$ A complete list of the metrics considered is provided in the Appendix E.

**Selective classification baselines**   To assess the efficacy of `SC-CE`, we compare it with two baseline state-of-the-art selective classifiers. The first is **PlugInRule** (Herbei and Wegkamp, 2006), which uses the predicted probabilities of the underlying classifier to fit a rejection threshold and **PlugInRuleAUC** (Pugnana and Ruggieri, 2023b), a variant of PlugInRule that uses the area under the ROC curve (AUC) as the performance metric to determine the optimal threshold for rejection, which improves performance in cases where the class distribution is imbalanced.

**Evaluation metrics**   The evaluation of an ML model with a reject option requires metrics that assess both the predictor and the rejection function: an ideal selective classifier should achieve high accuracy on non-rejected examples while maintaining a low rejection rate (Hendrickx et al., 2024), *i.e.*, should reject the misclassified samples. Several metrics exist to evaluate SC (Condessa et al., 2017), we employ Non-rejected Accuracy ($NA$), Classification Quality ($CQ$), and Rejection Quality ($RQ$) in our analysis. Following the terminology introduced in Table 1, they are defined as: $NA$ represents the accuracy on the subset of accepted samples, computed as the ratio of correct examples to total accepted instances. It is noteworthy that models with higher rejection rates may be favored when using this metric alone. $RQ$ represents the rejection policy's ability to reject misclassified

examples, computed by comparing the ratio of misclassified examples on the rejected subset with the ratio on the complete dataset. $CQ$ is an assessment of the classifier's performance on the set of non-rejected samples and the performance of the rejector on the set of misclassified samples. We report only the plots of $NA$ to illustrate the performance-rejection trade-off of our approach compared to baselines. The complete analysis for $NA, CQ$, and $RQ$, is provided in the Appendix H.

**Sampling strategy for the distance**  With bounded abstention, once a target coverage $C$ is defined, the $i$-th percentile of the ranked calibration set $X_{cal}$ is taken as a reference to set the right rejection threshold $\tau_r$. Because these distances inhabit a strictly non-negative space, often a right-skewed space shaped by the model's decision boundary, we experimented with our method not only through linear sampling but also through Gamma sampling, which we expect to reflect the empirical distribution of distances better, yielding lower-variance percentile estimates. To do so, we fit a Gamma distribution to the distances computed over the calibration set and then estimate the percentiles as for bounded abstention.

### 4.2 RESULTS

**RQ1 - Correlation with model's confidence**  We study the correlation between the instance-counterfactual distance computed on the calibration set of each dataset and the prediction probability of each classifier, the latter being a proper confidence metric for classification (Herbei and Wegkamp, 2006). Table 2 presents the results for the three most correlated distances, for all datasets and all models, with all counterfactual generation methods. From the table, we can see that generally using $\texttt{ILS}_{latent}$ yields good and fairly consistent correlations with $L_2$ distance, indicating that indeed a form of counterfactual distance can serve as a good proxy for the confidence of a classifier. Naturally, this heavily depends on the shape of the decision boundary of the classifier, as highlighted by the differences between the different models for the same dataset. The complete tables with the correlations between the instance-counterfactual distances metrics and the prediction probability for the $\texttt{Wisconsin}$ dataset (computed on the calibration and test set, grouped by counterfactual generation method, for every black box) are in the Appendix G.

| Dataset | Metric | LightGBM | | | | MLP | | | | Random Forest | | | | XGBoost | | | |
|---|---|---|---|---|---|---|---|---|---|---|---|---|---|---|---|---|---|
| | | DiCE | ILS | ILS$_{latent}$ | LoRE | DiCE | ILS | ILS$_{latent}$ | LoRE | DiCE | ILS | ILS$_{latent}$ | LoRE | DiCE | ILS | ILS$_{latent}$ | LoRE |
| Adult | Bray-Curtis | 0.74 | **0.95** | 0.93 | 0.64 | 0.78 | 0.86 | **0.93** | 0.70 | 0.75 | 0.90 | 0.89 | 0.70 | 0.67 | **0.94** | 0.90 | 0.64 |
| | Cosine | 0.64 | 0.91 | 0.93 | 0.50 | 0.71 | 0.85 | 0.91 | 0.52 | 0.70 | 0.86 | **0.95** | 0.57 | 0.58 | 0.91 | 0.93 | 0.49 |
| | $L_2$ | 0.33 | 0.68 | 0.90 | 0.46 | 0.35 | 0.23 | 0.73 | 0.42 | 0.30 | 0.04 | -0.05 | 0.48 | 0.29 | 0.79 | 0.90 | 0.46 |
| Wisconsin | Bray-Curtis | 0.37 | **0.73** | **0.73** | 0.29 | 0.31 | 0.42 | 0.42 | 0.39 | 0.52 | 0.78 | 0.78 | 0.67 | 0.41 | **0.73** | **0.73** | 0.47 |
| | Cosine | 0.21 | 0.72 | 0.72 | 0.27 | 0.17 | 0.53 | 0.53 | 0.39 | 0.28 | 0.73 | 0.73 | 0.58 | 0.22 | 0.69 | 0.69 | 0.46 |
| | $L_2$ | 0.20 | 0.53 | 0.53 | 0.31 | 0.21 | **0.58** | **0.58** | 0.55 | 0.34 | **0.82** | **0.82** | 0.63 | 0.26 | 0.58 | 0.58 | 0.54 |
| German | Bray-Curtis | 0.19 | 0.07 | **0.69** | 0.39 | 0.43 | -0.22 | 0.57 | **0.63** | -0.03 | 0.89 | 0.97 | 0.44 | -0.13 | 0.11 | **0.68** | -0.19 |
| | Cosine | 0.17 | 0.04 | 0.73 | 0.15 | 0.31 | -0.40 | 0.60 | 0.31 | -0.07 | 0.92 | 0.97 | 0.24 | -0.14 | 0.06 | 0.72 | -0.30 |
| | $L_2$ | 0.30 | -0.18 | 0.74 | 0.50 | 0.46 | -0.40 | 0.57 | 0.75 | 0.10 | 0.83 | **0.99** | 0.58 | 0.05 | 0.14 | 0.73 | 0.05 |
| Two-Moons | Bray-Curtis | 0.18 | 0.25 | -0.04 | -0.07 | 0.44 | -0.17 | -0.20 | 0.07 | 0.41 | 0.15 | 0.26 | -0.02 | 0.37 | 0.20 | 0.04 | 0.03 |
| | Cosine | -0.10 | 0.23 | 0.18 | 0.07 | -0.01 | 0.01 | **0.58** | -0.04 | -0.04 | 0.09 | 0.36 | -0.08 | 0.01 | 0.17 | 0.23 | 0.03 |
| | $L_2$ | **0.48** | 0.26 | -0.03 | 0.26 | 0.74 | -0.21 | -0.20 | 0.37 | **0.52** | 0.41 | 0.45 | 0.17 | **0.60** | 0.13 | 0.14 | 0.32 |

Table 2: Correlation between the top 3 distances (using $\min$ as aggregation function) and the prediction probability of each ML model on the calibration set.

**RQ2. $\texttt{SC-CE}$ performance against benchmarks**  The direct comparison of our method with the two baseline selective classification models is evaluated through the performance metrics introduced in Section 3: non-rejected accuracy $NA$ (see Figure 2) classification quality $CQ$ and rejection quality $RQ$. The complete results can be found in the Appendix H. Our results suggest that $\texttt{SC-CE}$ matches the performance or even exceeds the baselines, consistently for almost all target coverages. While in the case of $CQ$ all $\texttt{SC-CE}$ methods show similar performance, when considering the $NA$, we find that $\texttt{ILS}$ and $\texttt{ILS}_{latent}$ emerge as the preferred method to realize an effective rejection policy.

Our experiments systematically compared a wide range of counterfactual generation and rejection methods across several datasets and models. The Friedman test with $\alpha = 0.05$ has been used for the statistical analysis. The results show that, across all datasets and models, the best-performing rejection policy in terms of NR is the PlugInRule, followed by our proposal, specifically multiple combinations utilising $\texttt{LoRE}$ as the counterfactual generator method with various distance functions (e.g., mean absolute error, $L_1$ and Wasserstein metrics), all aggregated with the minimum. Likewise,

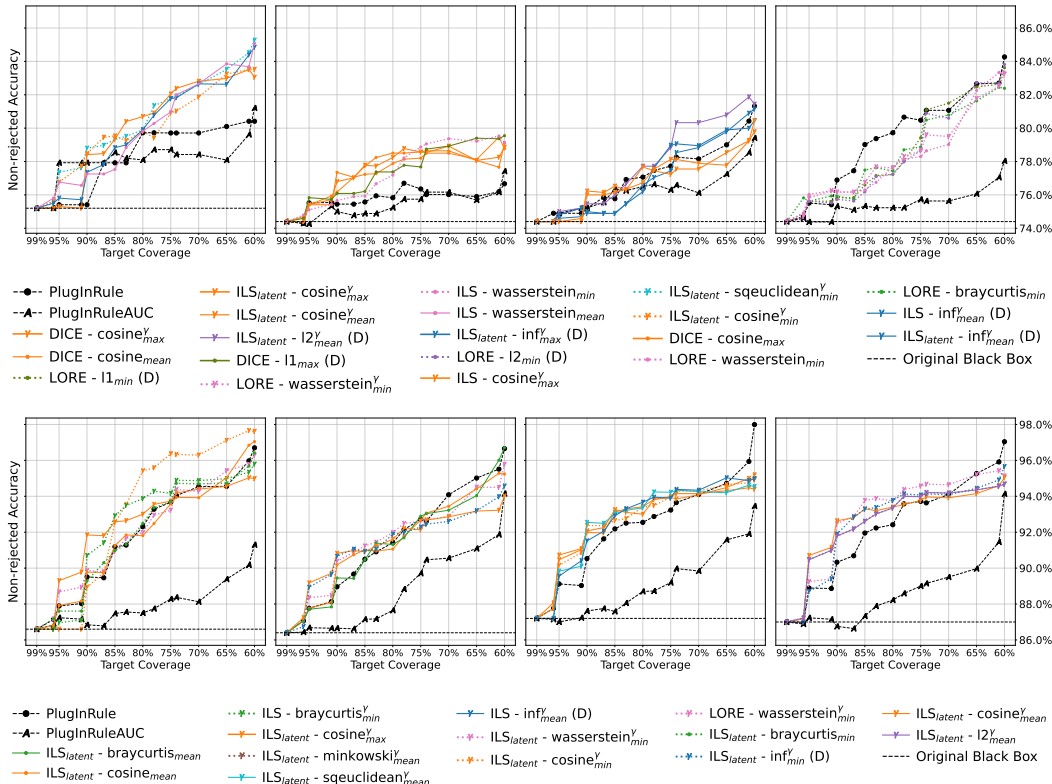

Figure 2: Non-Rejected Accuracy performance curves for the `German` and `Adult` dataset on the four classifiers: `RF`, `mlp`, `xgboost` and `LGBM`. In the plots, the $x$-axis indicates the target coverage, while on the $y$-axis we find the $NA$, with higher values indicating better performance. In the legend, we indicate with $\gamma$ symbols the `SC-CE` implementations where the percentiles are computed while fitting a Gamma distribution over the distances, while the two benchmarks are indicated with black lines. We present results only for the best 6 counterfactual methods for each combination of dataset and ML model.

when examining performance on each individual dataset, `SC-CE` always demonstrates competitive performance, particularly when using the counterfactual generation techniques $\texttt{ILS}_{latent}$ and `DiCE`. In all cases, the statistical comparison of selective classification performance indicates no statistically significant differences ($p$-value $> 0.05$) between the top-performing `SC-CE` combinations and the benchmark methods, confirming that our proposed method attains state-of-the-art performance while providing the additional benefit of interpretable explanations. The full critical difference diagrams are available in Appendix F.

**RQ3. `SC-CE` explanations** Figure 16 displays an example of the explanation provided by `SC-CE` in the case of a rejected instance selected from the `Adult` dataset. In this case, the method refrains from making a prediction due to uncertainty near the decision boundary and instead provides a multimodal explanation for the decision to abstain. By employing the explanatory format delineated in Section 3, `SC-CE` aims to clarify to a prospective user of the system that no reliable prediction can be rendered, as the input lies within an uncertainty region of the ML model. To support this claim, it shows that opposite outcomes may be achieved with only marginal modifications to the input, thereby invalidating any prediction.

Furthermore, Figure 16 in Appendix I presents an example of output provided by `SC-CE` for an accepted test instance of the `Adult` dataset. This output contains a multi-modal explanation analogous to the rejection scenario in addition to the ML prediction.

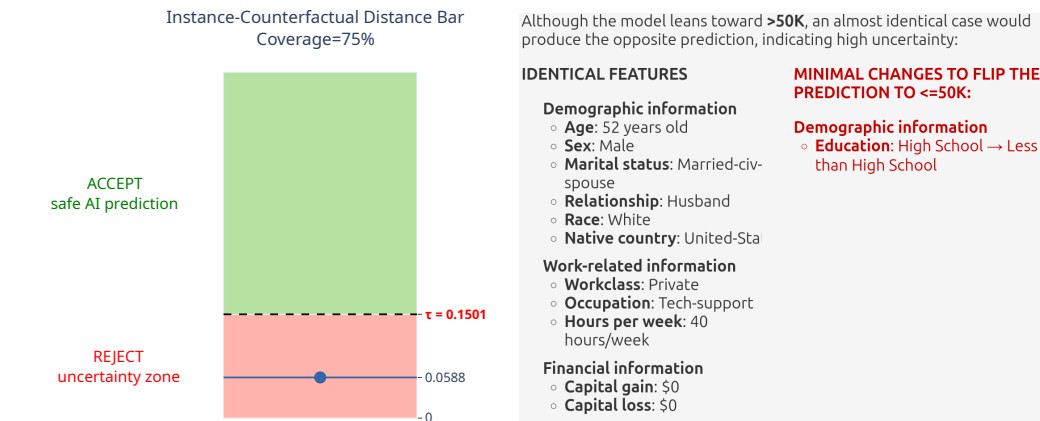

Figure 3: Example of explanation generated by `SC-CE` for a *rejected* test instance of the `Adult` dataset. The output includes a banner warning the user that the AI system cannot provide a prediction for the input due to its placement inside the model's uncertainty zone, along with instructions to properly interpret the figures.

## 5 DISCUSSION AND CONCLUSION

We have introduced *Selective Classification via Counterfactual Explanations* (`SC-CE`), a novel framework for selective classification that implements contrastive and model-agnostic explanation of the reject option using only hard predictions. `SC-CE` employs a rejection policy that considers the distance between input data and their counterfactuals as a proxy for model confidence. `SC-CE` is thus able to provide a multi-modal counterfactual explanation to clarify what is contributing to the classifier's uncertainty. The experimental results demonstrate that the distance between an input and its counterfactual explanations provides a viable proxy for model confidence, confirming our hypothesis that "easy" counterfactuals (i.e., small perturbations) correspond to low confidence, and vice versa. In terms of selective classification performance, `SC-CE` matches or surpasses the non-rejected accuracy of state-of-the-art baselines (PlugInRule, PlugInRuleAUC) across both the German and Adult datasets and all four learning models. Notably, employing a Gamma-fitted percentile ($\gamma$) often reduces variance at extreme coverage targets, yielding smoother accuracy-coverage curves than linear sampling of empirical order statistics. In addition to quantitative performance, `SC-CE` offers a human-readable explanation for rejection by directly employing the estimation of counterfactuals. This transparency directly addresses the opacity of conventional rejectors and enhances user trust in high-stakes decision-making scenarios.

### 5.1 LIMITATIONS AND FUTURE WORKS

The generation of counterfactuals requires non-trivial computational overhead. The use of a single global threshold $\tau$ rather than region-specific ones may overlook local heterogeneity in the feature space. The instance-counterfactual distance is primarily a measure of *boundary proximity* (ambiguity) and does not reliably detect novelties or out-of-distribution (OOD) inputs. We therefore recommend combining `SC-CE` with an explicit OOD/novelty detector (e.g., density estimators, reconstruction error, or a conformal novelty procedure) when novelty detection is required. Nonetheless, the modularity of `SC-CE` facilitates extensions such as integrating a deferral policy (Mozannar et al., 2023) that routes ambiguous cases to specific human experts, exploring the abstention policy for anomaly and novelty detection, possibly leveraging the cardinality of the set of admissible labels in close proximity to the input data through conformal prediction (Hallberg Szabadváry et al., 2025), expanding the explanation with guidance on how to turn a rejected instance into an accepted one, and conducting user studies to assess the subjective clarity and utility of counterfactual explanations in real-world decision workflows. Our framework, further strengthened by these advancements, provides a practical approach to reliable human-AI collaboration where ML predictions are supported by both abstention mechanisms and interpretable features.

## REPRODUCIBILITY STATEMENT

The anonymized code repository can be found and downloaded at: `https://anonymous.4open.science/r/L2R-CE_/Readme.md`. A description of the hardware, classification, and counterfactual generation methods, training process, and distance functions employed in the experiments is provided in the Appendix, along with additional numerical and graphical results for each research question.

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

APPENDIX

## A CONFORMAL PREDICTION FORMULATION

When a single canonical counterfactual $E(x) = x'$ is returned, `SC-CE` scores can be embedded in a split-conformal calibration to obtain distribution-free guarantees. Define a nonconformity score on the calibration set as

$$\alpha_i = S(x_i) = \frac{1}{d(x_i, E(x_i))}, \qquad i = 1, \ldots, n_{\text{cal}},$$

so that larger $\alpha$ indicates closer counterfactuals (higher nonconformity/uncertainty). For a target coverage level $\delta \in (0, 1)$ compute the $(1 - \delta)$-quantile $q_{1-\delta}$ of the calibration scores $\{\alpha_i\}$ using the usual conformal quantile, i.e. the $\lceil (1 - \delta)(n_{\text{cal}} + 1) \rceil$-th smallest value. For a test point $x$ define $\alpha_{\text{test}} = S(x)$. A conservative rejection rule is

$$\text{reject } x \iff \alpha_{\text{test}} > q_{1-\delta}.$$

Equivalently, the conformal $p$-value may be computed as

$$p(x) = \frac{|\{i : \alpha_i \geq \alpha_{\text{test}}\}| + 1}{n_{\text{cal}} + 1},$$

and the instance is accepted only when $p(x) > \delta$. This follows the standard split-conformal calibration procedure.

## B HARDWARE

All experiments were conducted on a high-performance workstation running Ubuntu 22.04.5 LTS, equipped with an Intel® Core™ i9-10980XE CPU @ 3.00GHz (18 cores, 36 threads), 251 GB of memory, and two NVIDIA GPUs: a Quadro RTX 6000 for compute-intensive tasks and a GeForce GT 1030 for display. The system was configured with NVIDIA driver version 535.247.01.

## C DATASET DESCRIPTION

| Dataset | Features | Samples | Class Ratio |
|---|---|---|---|
| Two-Moons | 2 | 1000 | 50:50 |
| Adult | 12 | 48842 | 76:24 |
| German | 20 | 1000 | 70:30 |
| Wisconsin | 30 | 569 | 63:37 |

Table 3: The Datasets considered for the experiments are tabular data.

## D HYPERPARAMETER OPTIMIZATION

**Black box classifiers** The hyperparameter is determined by doing a cross-validation on the training set with 5 folds. The best hyperparameters are selected based on the average accuracy across the folds. The hyperparameter search is done using a random search approach, where some combinations of the specified options are evaluated. While not of great importance, the hyperparameter search is done to ensure that the models are not overfitting or underfitting the data. The hyperparameters and their possible values, specific for each black box model, can be found in Table 4.

In the Table 5, there are the best hyperparameters selected for each model and dataset. The hyperparameters are chosen based on the average accuracy across the folds.

**Counterfactual Generators: `DiCE` and `LoRE`** We used fixed hyperparameters for the generation of the counterfactuals when using `DiCE` and `LoRE`. For `DiCE` we used the proximity weight equal to 0.5 and as a method `kdtree`. For `LoRE` we used the `genetic` option for the neighborhood synthesis.

| Model | Hyperparameter | Values |
|---|---|---|
| **RF** | num estimators | 50, 100, 150 |
| | max depth | 6, 10, 30, None |
| | min samples split | 2, 5, 10 |
| | min samples leaf | 1, 2, 4 |
| **mlp** | num hidden layers | 1, 2 |
| | hidden neurons | 32, 64, (32, 32), (64, 32) |
| | learning rate | 0.001, 0.01, 0.1 |
| | max epochs | 150, 300, 500 |
| **xgboost** | num estimators | 50, 100, 150 |
| | max depth | 3, 6, 9, None |
| | learning rate | 0.01, 0.05, 0.1 |
| | min child weight | 1, 3, 5 |
| **LGBM** | num estimators | 50, 100, 150 |
| | max depth | 6, 10, 30, None |
| | learning rate | 0.01, 0.05, 0.1 |

Table 4: Hyperparameter configurations for different models, including `RF`, `mlp`, `xgboost`, and `LGBM`. The choices for each hyperparameter are specified.

| Model | Hyperparameters | German | Adult | Wisconsin | Two-Moons |
|---|---|---|---|---|---|
| **RF** | num estimators | 50 | 150 | 50 | 50 |
| | max depth | 10 | 30 | 10 | 10 |
| | min samples split | 2 | 5 | 5 | 5 |
| | min samples leaf | 1 | 4 | 1 | 1 |
| **mlp** | hidden layers | 1 | 1 | 1 | 1 |
| | hidden neurons | 64 | 32 | 64 | 32 |
| | learning rate | 0.001 | 0.01 | 0.01 | 0.1 |
| | max epochs | 150 | 300 | 300 | 500 |
| **xgboost** | num estimators | 150 | 150 | 150 | 50 |
| | max depth | 6 | 6 | 9 | None |
| | learning rate | 0.1 | 0.1 | 0.1 | 0.01 |
| | min child weight | 1 | 1 | 3 | 3 |
| **LGBM** | num estimators | 100 | 100 | 100 | 50 |
| | max depth | 10 | 30 | None | 30 |
| | learning rate | 0.05 | 0.05 | 0.05 | 0.01 |

Table 5: Best hyperparameters identified for each model and dataset through our optimization process.

**Counterfactual Generators: `ILS`**   `ILS` is optimized using a hyperparameter search using a stratified 3-fold random search, looking for 20 different random combinations within the hyperparameter space composed of:

- latent space dimensions: 2, 3, 4,

- batch size: 4, 8, 16, 32, 64, 128,

- learning rate: 0.0001, 0.001, 0.005, 0.008, 0.01

- sigma: 0.5, 1.0, 2.0

for a maximum of 2000 epochs with an early stop of 50 for the `German`, `Two-Moons`, and `Wisconsin` datasets, while 70 for `Adult`. What is optimized during the hyperparameter search is the KL-loss function of `ILS`, on the calibration set. We found that the `ILS`' best hyperparameters do not vary when varying the black-box model, but they remain the same across the dataset. For the `German` we found the best latent dimension to be 4, the best batch size to be 4, the learning rate equal to 0.0001, and $\sigma$=0.5; for the `Adult` dataset and `Wisconsin`, the best hyperparameters remained the same. `ILS` represent the space $(\mathcal{X}, \mathcal{Y})$ in an analogous way, performing best when

the number of dimensions is higher, while approaching the solution very slowly, having the lowest possible learning rate and the lowest batch size as well.

We do report that a different combination was obtained for the `Two-Moons`, which was indeed different: latent dimension=3, batch size=16, learning rate=0.001, and $\sigma$=1.0.

## E    DISTANCE FUNCTIONS

**Distance metrics**   A central element of our approach is quantifying the relationship between an instance and its counterfactuals. The choice of distance metric significantly impacts how this relationship is measured, as different metrics capture different aspects of the feature space. We explore a diverse set of distance functions to identify which best correlates with model confidence.

Given an instance $x$ to be selectively classified and a counterfactual $c$ in its set of counterfactuals $X_c$, we compute distances using multiple metrics. After preliminary experiments, we selected the following subset for our final analysis:

- **Standard norm-based distances:**
    - $L_1$ **(Manhattan):** $d(x,c) = \|x - c\|_1 = \sum_i |x_i - c_i|$
    - $L_2$ **(Euclidean):** $d(x,c) = \|x - c\|_2 = \sqrt{\sum_i (x_i - c_i)^2}$
    - **Squared Euclidean:** $d(x,c) = \sum_i (x_i - c_i)^2$
    - **Chebyshev ($L_\infty$):** $d(x,c) = \|x - c\|_\infty = \max_i |x_i - c_i|$
- **Similarity-based measures:**
    - **Cosine distance:** $d(x,c) = 1 - \frac{x \cdot c}{\|x\|\|c\|}$
    - **Bray-Curtis distance:** $d(x,c) = \frac{\sum_i |x_i - c_i|}{\sum_i (|x_i| + |c_i|)}$
- **Other specialized distances:**
    - **MAE (Mean Absolute Error):** $d(x,c) = \frac{1}{n} \sum_i |x_i - c_i|$
    - **Minkowski:** $d(x,c) = \left(\sum_i |x_i - c_i|^p\right)^{\frac{1}{p}}$
    - **Wasserstein:** A statistical distance between probability distributions

The diversity of these metrics allows us to thoroughly investigate our hypothesis that distance from counterfactuals serves as a proxy for model confidence. Each metric offers a different perspective: norm-based distances measure absolute differences in feature space; similarity measures capture orientation differences regardless of magnitude; while specialized distances like Wasserstein can better handle distributional shifts. For instance, $L_1$ is less sensitive to outliers than $L_2$, while cosine distance focuses on angle rather than magnitude, which is valuable for high-dimensional spaces.

This comprehensive evaluation provides insights into which geometric properties of the instance-counterfactual relationship best indicate the reliability of model predictions, directly addressing our first research question (RQ1). The full tables reporting the correlations between the confidence of the classification model and the instance-counterfactual distance, for all considered distance functions, are displayed in Section G for the dataset `Wisconsin`.

## F    CRITICAL DIFFERENCE DIAGRAMS

Here we report the critical difference diagrams. We focus our statistical analysis on Non-Rejected Accuracy as it captures the primary objective of rejection learning: prediction quality on retained instances. Evaluating the quality of counterfactual explanations would require either user studies or XAI metrics that are known in the literature to lack robustness.

The critical difference diagrams provide a comprehensive view of method performance across different datasets and experimental conditions. Each method name follows the format: `[Counterfactual Generator] - [Distance Function][Aggregation Method]`, where counterfactual generation includes the ones described in the paper, the distance functions are those listed in E, and aggregation methods include min, max, and mean.

For each test, the methods are ranked by mean performance across target coverage levels ranging from 50% to 99%. Than a Friedman test is performed using a significance level of $\alpha = 0.05$. The critical difference (CD) is computed based on the number of methods and datasets, and methods that are not significantly different are connected with a horizontal line. The higher the rank (leftmost in the plot), the better the performance of the method.

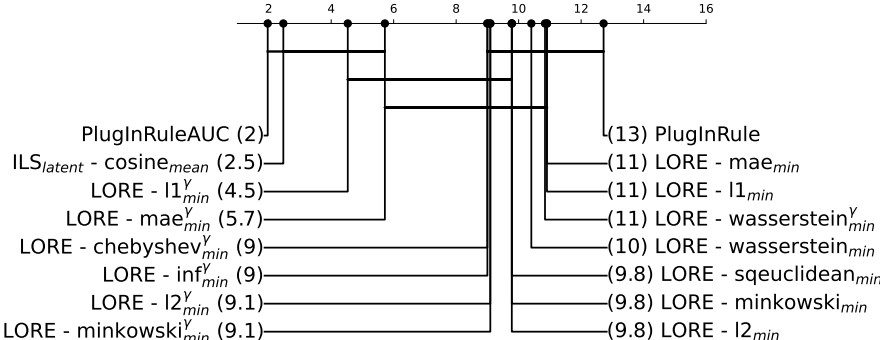

Figure 4: Critical difference diagram showing the best-performing rejection policies in terms of Non-Rejected Accuracy (NA) across all four datasets and models. PlugInRuleAUC achieves the best overall performance, followed by $\texttt{ILS}_{latent}$ with cosine distance and mean aggregation. The absence of connecting lines indicates no statistically significant differences between top-performing methods.

## F.1 DATASET-SPECIFIC ANALYSIS

Figure 5 shows a pattern for $\texttt{Adult}$: counterfactual-based methods show competitive performance alongside traditional approaches. Given the normalized feature space with target-encoded categorical variables, the prominence of $\texttt{DiCE}$, $\texttt{LoRE}$ methods with Bray-Curtis distance suggests this metric effectively captures relationships in the transformed feature representation. $\texttt{ILS}$ appear in the best performing when paired with Bray-Curtis distance (rank 7.6). The relatively high $p$-value indicates that while there are performance differences, they may not be statistically robust, highlighting the competitive nature of multiple approaches when working with standardized feature spaces.

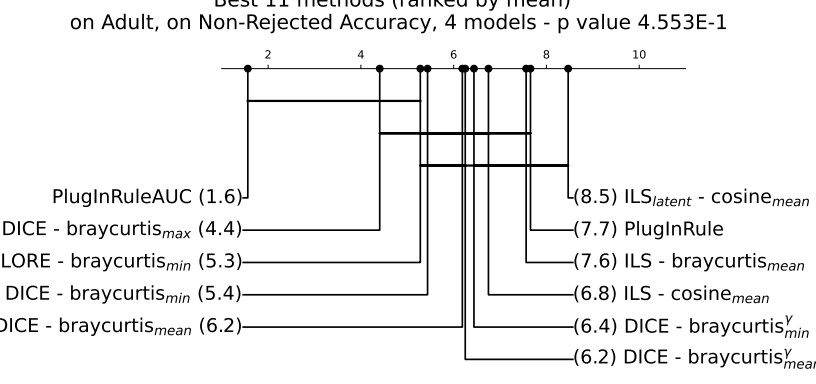

Figure 5: Critical difference diagram for the $\texttt{Adult}$ dataset showing rejection policy performance across 4 models. PlugInRuleAUC maintains its leading position with a mean rank of 1.6, followed by $\texttt{DiCE}$-based rejection policy using Bray-Curtis distance. $\texttt{DiCE}$ methods perform competitively.

The $\texttt{German}$ dataset results reveal a dataset where counterfactual-based rejection policies show their strongest relative performance. The dominance of $\texttt{LoRE}$ methods with Wasserstein distance

is particularly noteworthy, as this suggests that optimal transport distances are well-suited to the normalized feature space created by target encoding. The methods with the equivalent ranking ling in the second half of the CD diagram, in Figure 6 can be verified to reject the same instances or, anyway, maintain the same non-rejected accuracy.

Best 11 methods (ranked by mean)
on German, on Non-Rejected Accuracy, 4 models - No significant difference

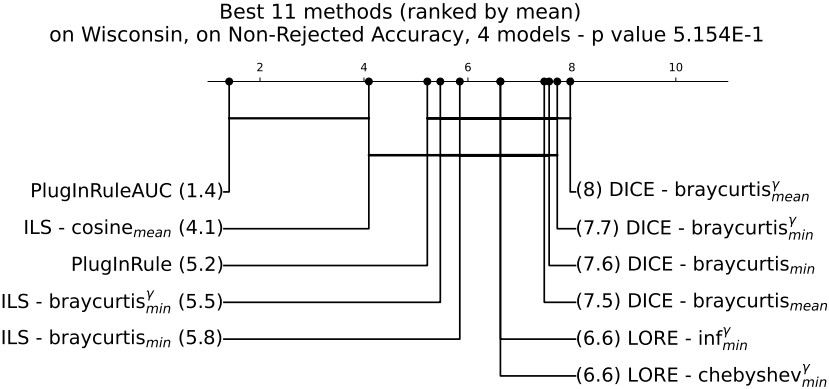

Figure 6: Critical difference diagram for the `German` dataset across 4 models. `LoRE` methods with Wasserstein distance with min aggregation achieve the top rankings, demonstrating the effectiveness of optimal transport-based distances for this social-economic dataset. PlugInRule methods maintain competitive performance but are outranked by the counterfactual-based approaches.

The `Wisconsin` dataset results demonstrate another scenario where the proposed `ILS` method achieves strong performance, ranking second after the traditional PlugInRuleAUC. The effectiveness of `ILS` with cosine distance in the normalized feature space suggests this combination is strong. The presence of multiple `ILS` variants in the top tier in Figure 7 indicates robustness across different distance functions, while the relatively high $p$-value confirms the competitive nature of multiple approaches in this domain.

Best 11 methods (ranked by mean)
on Wisconsin, on Non-Rejected Accuracy, 4 models - p value 5.154E-1

Figure 7: Critical difference diagram for the `Wisconsin` dataset across 4 models. PlugInRuleAUC maintains its leading position with a mean rank of 1.4, followed by `ILS` with cosine distance and mean aggregation (rank 4.1). The dataset shows strong performance for `ILS` methods with both cosine and Bray-Curtis distances, while `DiCE` and `LoRE` methods occupy middle-tier positions.

The `Two-Moons` results provide insights into method behavior on synthetic data with well-defined decision boundaries. The strong performance of traditional plug-in methods aligns with expectations for datasets where the underlying structure is relatively simple and geometric. The success of `DiCE` methods with standard distance metrics (L1, L2) further supports the idea that synthetic datasets with clear separability favor straightforward distance-based rejection strategies over more complex counterfactual generation approaches.

**Best 12 methods (ranked by mean)**
**on Two-Moons, on Non-Rejected Accuracy, 4 models - No significant difference**

PlugInRuleAUC (1.3)

PlugInRule (3)

DICE - $l1^\gamma_{mean}$ (6.3)

DICE - $l1^\gamma_{max}$ (6.9)

DICE - $mae^\gamma_{max}$ (6.9)

DICE - $l2^\gamma_{mean}$ (6.9)

(8.1) DICE - $inf^\gamma_{max}$

(8.1) DICE - $chebyshev^\gamma_{max}$

(7.8) DICE - $wasserstein^\gamma_{max}$

(7.8) DICE - $minkowski^\gamma_{max}$

(7.8) DICE - $l2^\gamma_{max}$

(6.9) DICE - $minkowski^\gamma_{mean}$

Figure 8: Critical difference diagram for the synthetic `Two-Moons` dataset across 4 models. PlugIn-Rule methods secure the top two positions, with `DiCE` methods using various distance functions (L1, L2, and others) filling out the remaining top-tier positions. The clear performance hierarchy suggests that simpler geometric relationships in synthetic data favor more direct rejection approaches.

Proposal's performance varies across datasets despite consistent methodology. Different rejection strategies show strengths on different data types, suggesting method selection should be data-driven rather than universal. Minimum aggregation dominates in `LoRE` methods, while mean aggregation works well for `ILS` methods, indicating that different counterfactual generators may benefit from different aggregation approaches. The absence of statistically significant differences among top-performing methods, as indicated by the connecting lines in the critical difference diagrams, suggests that multiple approaches can achieve comparable performance; the choice of method may depend on other factors like computational efficiency or interpretability requirements; the proposed counterfactual-based approaches are competitive with established baselines.

## G    ADDITIONAL RESULTS FOR RQ1: CORRELATION WITH MODEL'S CONFIDENCE

The following tables report the correlation between the instance-counterfactual distance metrics computed on the calibration set of the `Wisconsin` dataset and the prediction probability of each classification method, split by counterfactual generation method.

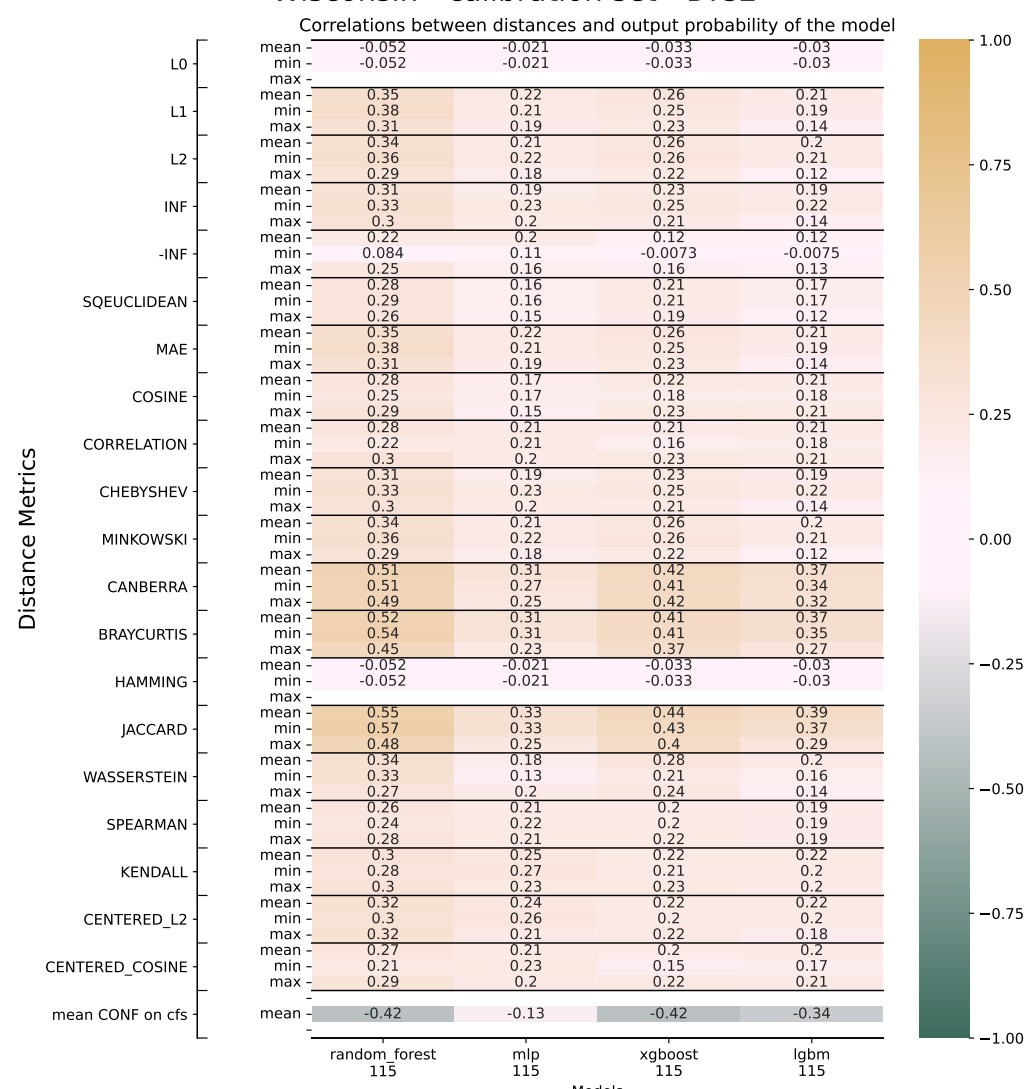

Figure 9: Distance-confidence correlation for the dataset `Wisconsin` and the counterfactual generation method `DiCE`

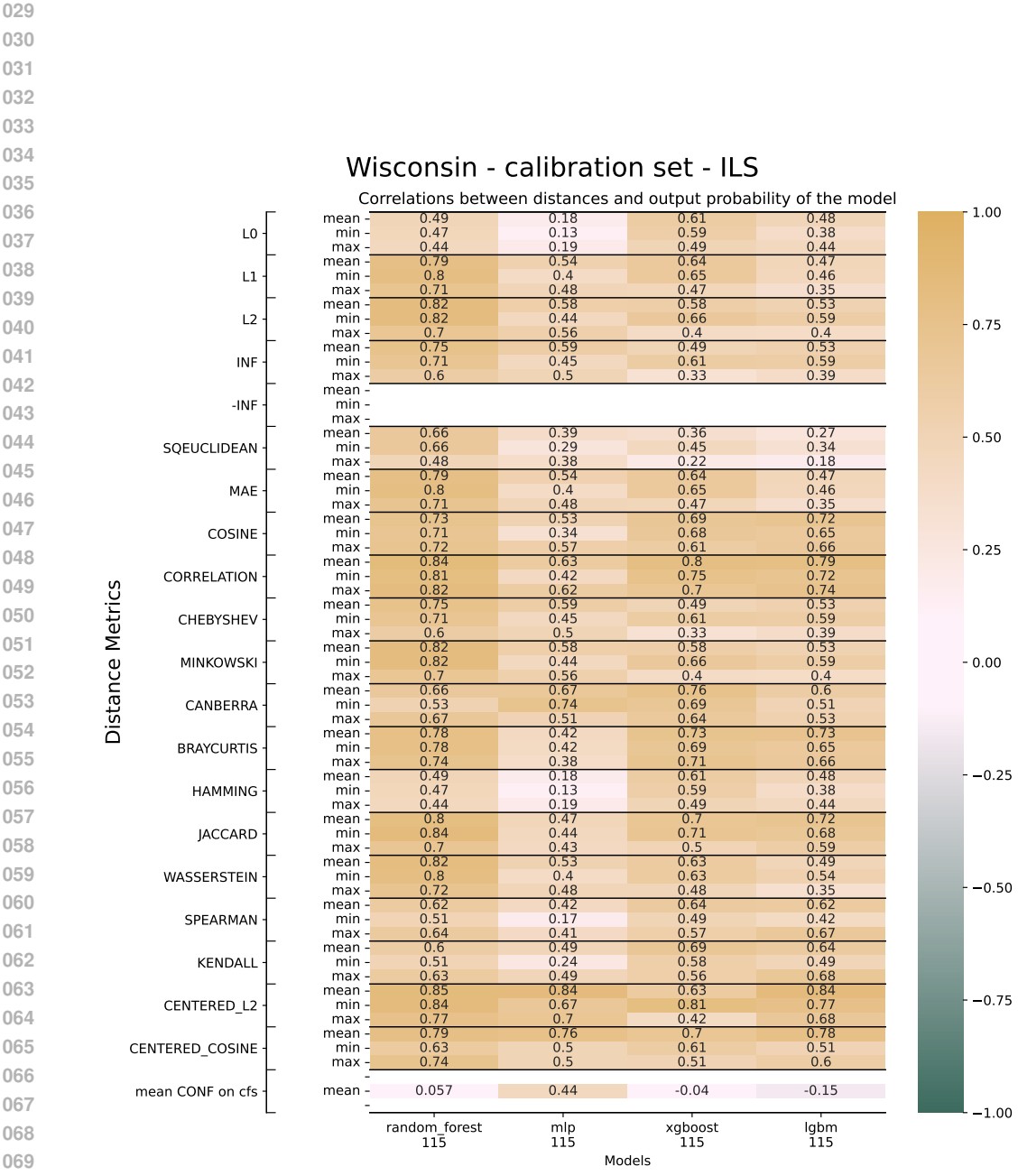

Figure 10: Distance-confidence correlation for the dataset `Wisconsin` and the counterfactual generation method `ILS`

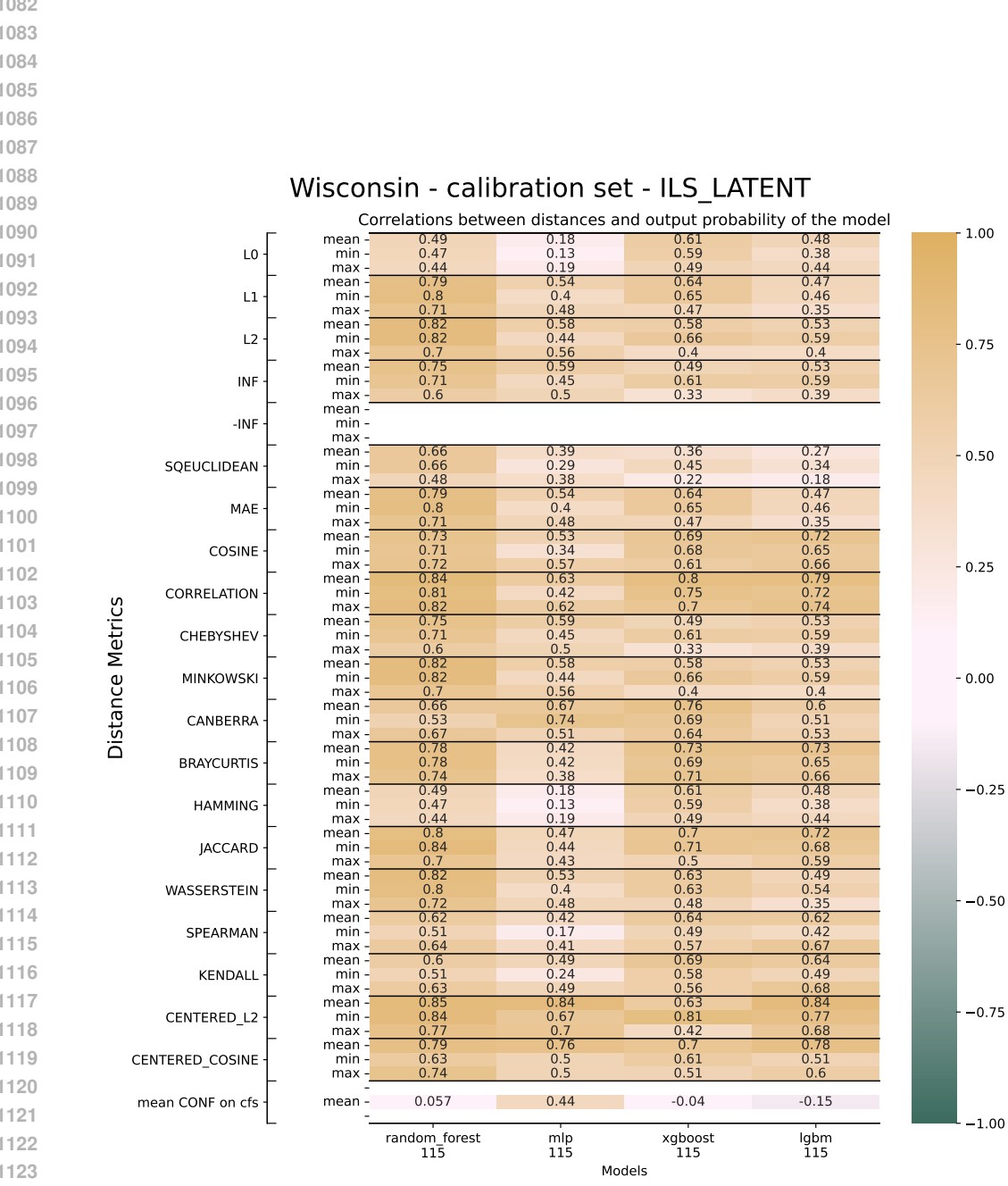

Figure 11: Distance-confidence correlation for the dataset `Wisconsin` and the counterfactual generation method $ILS_{latent}$

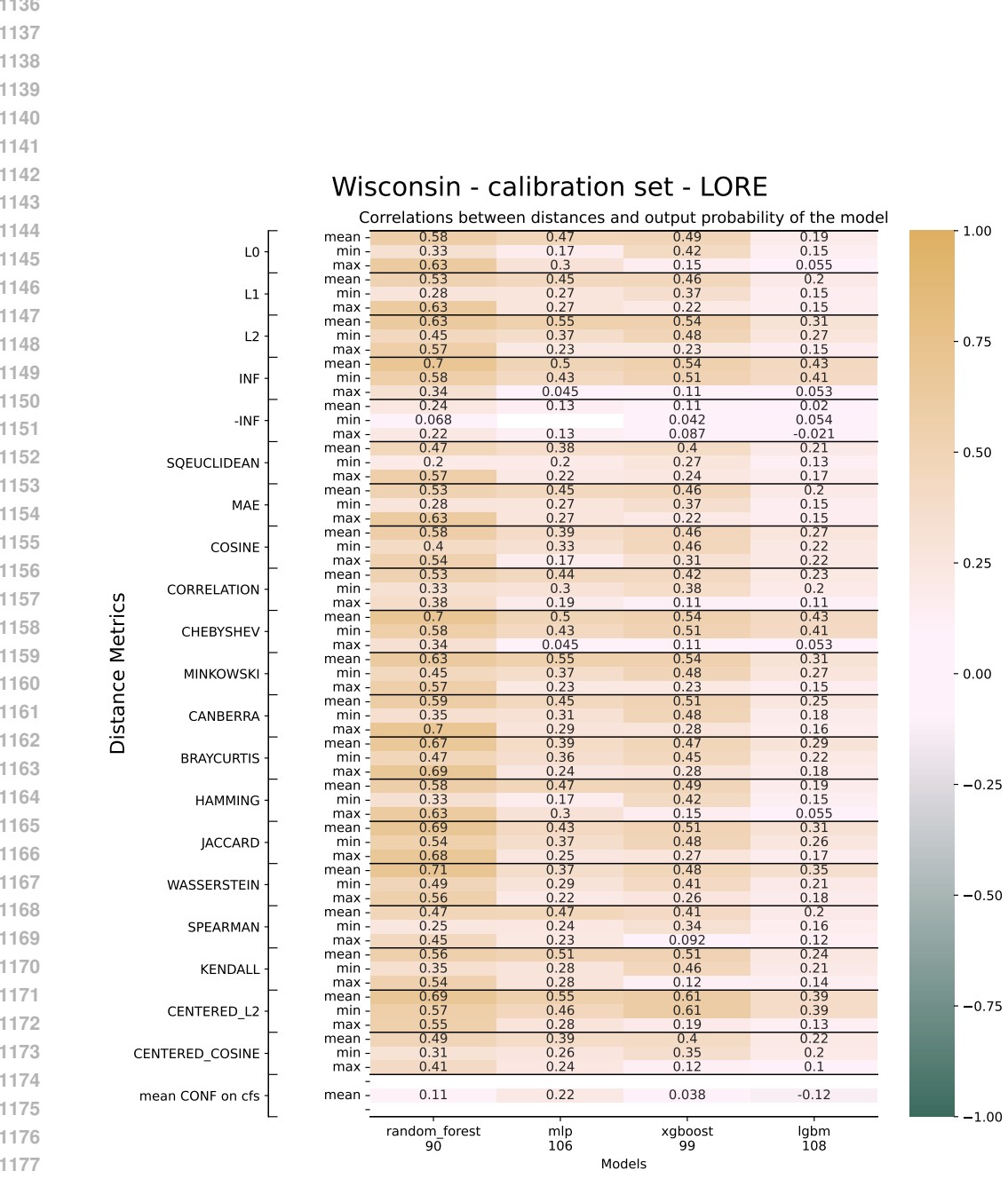

Figure 12: Distance-confidence correlation for the dataset `Wisconsin` and the counterfactual generation method `LoRE`

# H ADDITIONAL RESULTS FOR RQ2: SC-CE PERFORMANCE AGAINST BENCHMARKS

**Additional plots** In this section, we include additional plots displaying our results regarding the performance of top SC-CE combinations against the two chosen selective classification baselines, PlugInRule and PlugInRuleAUC on the benchmark datasets. Performance is computed as a function of the target coverage. Specifically, we report:

- the Non-rejected Accuracy $NA$ of Wisconsin and Two-Moons in Figure 13;
- the Classification Quality $CQ$ of Wisconsin and Two-Moons in Figure 14;
- the Classification Quality $CQ$ of German and Adult in Figure 15;

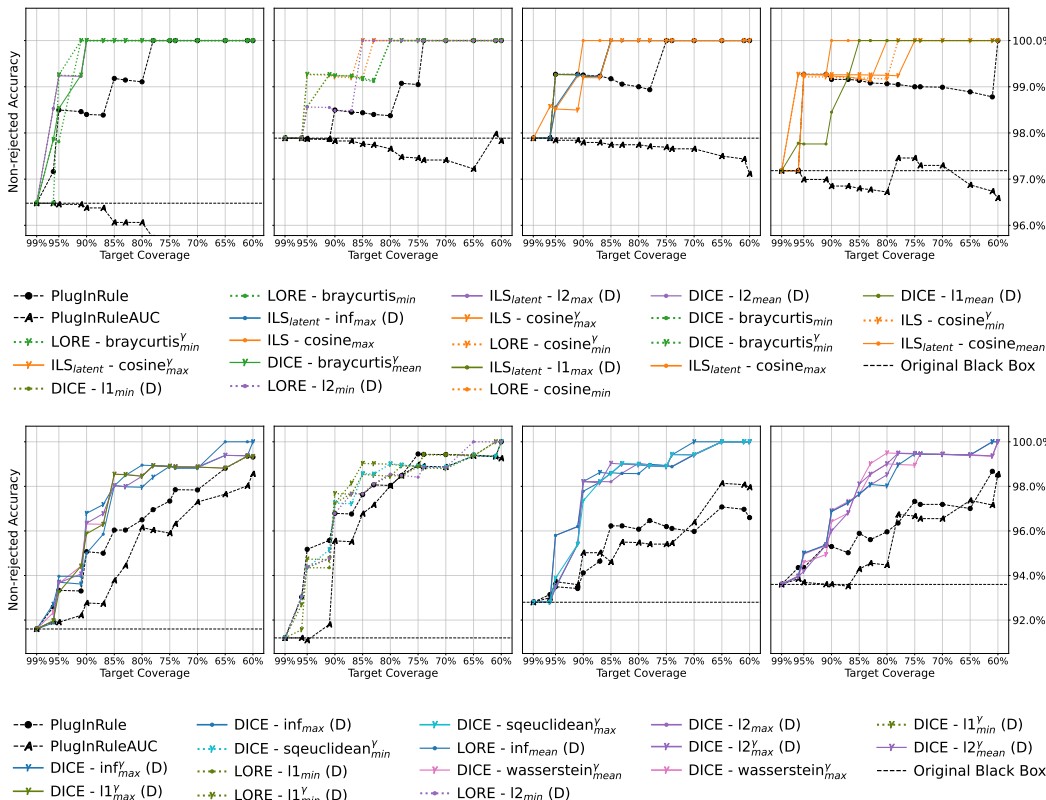

Figure 13: Non-Rejected Accuracy performance curves for the Wisconsin (top row) and Two-Moons (bottom row) dataset on the four classifiers: RF, mlp, xgboost and LGBM. In the plots, the $x$-axis indicates the target coverage, while on the $y$-axis we find the $NA$, with higher values indicating better performance. In the legend, we indicate with $\gamma$ symbols the SC-CE implementations where the percentiles are computed while fitting a Gamma distribution over the distances, while the two benchmarks are indicated with black lines. We present results only for the best 6 counterfactual methods for each combination of dataset and ML model.

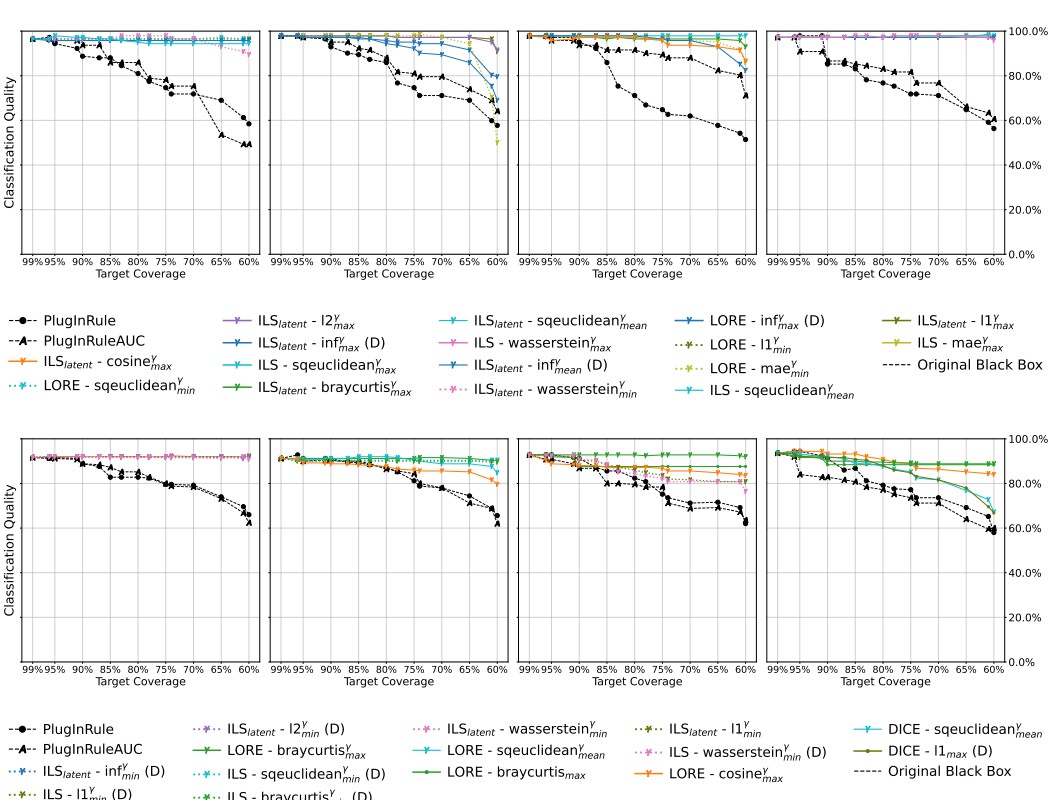

Figure 14: Classification Quality metrics for the `Wisconsin` (top row) and `Two-Moons` (bottom row) dataset. The upper plot shows performance curves. In the plots, the $x$-axis indicates the target coverage while on the $y$-axis we find the $NA$ or $CQ$ metric, which in both cases ranges between 0 and 1, with higher values indicating better performance. In the legend of the figures, we indicate with $\gamma$ symbols the `SC-CE` implementations where the percentiles are computed while fitting a Gamma distribution over the distances, while the two benchmarks are indicated with black lines. We present results only for the best 6 counterfactual methods for each combination of dataset and ML model. In the first plot, there is the `RF`, the second is the `mlp`, the third is the `xgboost`, and the last is the `LGBM`.

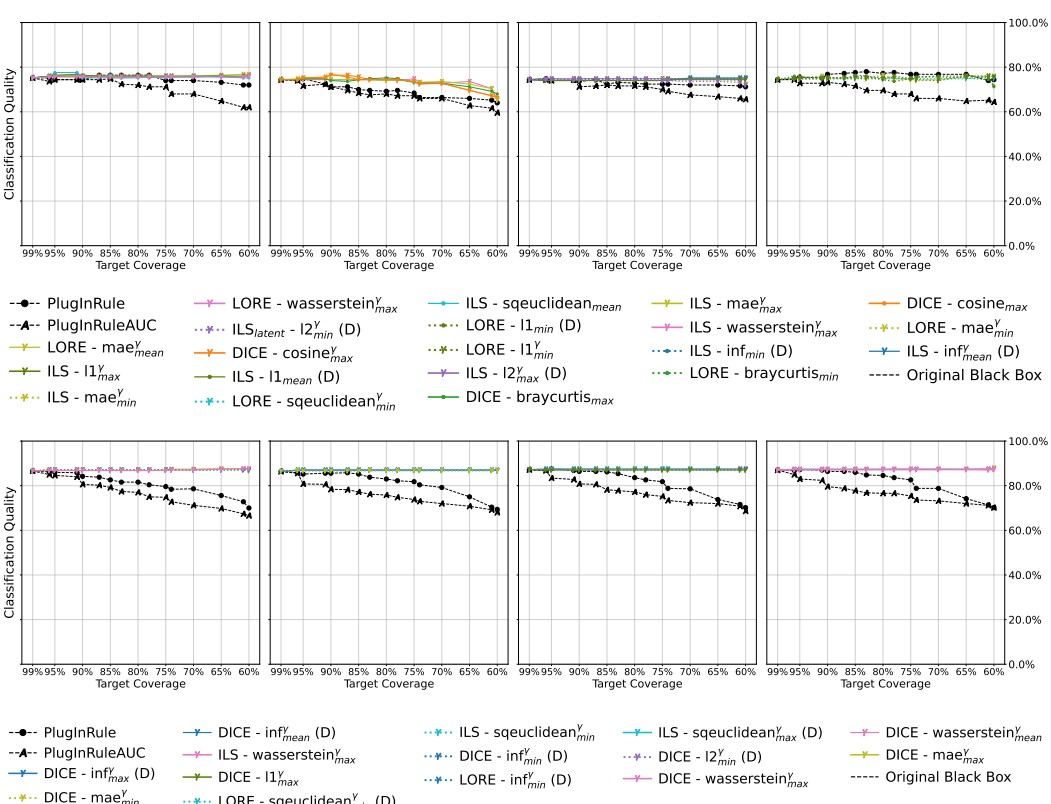

Figure 15: Classification Quality performance curves for the `German` (top row) and `Adult` dataset (bottom row) on the four classifiers: `RF`, `mlp`, `xgboost` and `LGBM`. In each plots, the $x$-axis indicates the target coverage while on the $y$-axis we find the $CQ$, with higher values indicating better performance. In the legend, we indicate with $\gamma$ symbols the `SC-CE` implementations where the percentiles are computed while fitting a Gamma distribution over the distances, while the two benchmarks are indicated with black lines. We present results only for the best 6 counterfactual methods for each combination of dataset and ML model.

**Full tables** The following tables report the performance of SC-CE and the two chosen selective classification baselines on all benchmark datasets. Performance is computed in terms of Non-rejected Accuracy, Classification Quality, and Rejection Quality. As described in the experimental setting (4), SC-CE tested on all combinations of the selected counterfactual explainers $E$ (i.e., LoRE, DiCE, ILS and $ILS_{latent}$), distance metric $d$ (e.g., Euclidean, Manhattan, cosine, etc.) and aggregation policy $a$ (i.e., minimum, maximum and mean). We limit the content of the following tables only to the top performing combinations for each dataset D and black-box classifier $f$ pair. Specifically, we report:

- the Non-rejected Accuracy $NA$ of German and Adult in Table 6;
- the Non-rejected Accuracy $NA$ of Wisconsin and Two-Moons in Table 7;
- the Classification Quality $CQ$ of German and Adult in Table 8;
- the Classification Quality $CQ$ of Wisconsin and Two-Moons in Table 9;
- the Rejection Quality $RQ$ of German and Adult in Table 10;
- the Rejection Quality $RQ$ of Wisconsin and Two-Moons in Table 11.

| Dataset | Black Box | Rejection Policy | Target Coverages | | | | | | | | |
|---|---|---|---|---|---|---|---|---|---|---|---|
| | | | 99% | 95% | 90% | 85% | 80% | 75% | 70% | 65% | 60% |
| German | RF | PlugInRule | **0.752** | 0.754 | 0.754 | 0.779 | 0.797 | 0.797 | 0.797 | 0.801 | 0.804 |
| | | PlugInRuleAUC | **0.752** | **0.779** | 0.779 | 0.785 | 0.781 | 0.787 | 0.784 | 0.781 | 0.812 |
| | | $ILS_{latent}$ - $sqeuclidean^{\gamma}_{min}$ | **0.752** | 0.774 | **0.788** | 0.793 | 0.799 | 0.818 | 0.826 | 0.835 | **0.853** |
| | | $ILS$ - $cosine^{\gamma}_{max}$ | **0.752** | 0.753 | 0.784 | 0.793 | **0.807** | **0.821** | **0.828** | 0.830 | 0.831 |
| | | $ILS_{latent}$ - $cosine^{\gamma}_{min}$ | **0.752** | 0.768 | 0.785 | **0.795** | 0.798 | 0.809 | 0.819 | 0.832 | 0.835 |
| | | $ILS$ - $inf^{\gamma}_{mean}$ | **0.752** | 0.758 | 0.774 | 0.788 | 0.799 | 0.818 | 0.827 | 0.826 | 0.848 |
| | | $ILS$ - $wasserstein_{mean}$ | **0.752** | 0.768 | 0.773 | 0.775 | 0.798 | 0.810 | 0.827 | **0.839** | 0.851 |
| | mlp | PlugInRule | **0.744** | 0.755 | 0.754 | 0.756 | 0.758 | 0.764 | 0.760 | 0.759 | 0.767 |
| | | PlugInRuleAUC | **0.744** | 0.743 | 0.750 | 0.749 | 0.752 | 0.757 | 0.762 | 0.757 | 0.774 |
| | | $DICE$ - $cosine_{max}$ | **0.744** | 0.754 | **0.774** | 0.777 | **0.785** | 0.786 | 0.786 | 0.781 | 0.791 |
| | | $DICE$ - $cosine^{\gamma}_{max}$ | **0.744** | 0.754 | 0.768 | **0.778** | 0.782 | 0.786 | 0.789 | 0.781 | 0.789 |
| | | $DICE$ - $cosine_{mean}$ | **0.744** | 0.754 | 0.762 | 0.772 | 0.779 | 0.782 | 0.785 | 0.781 | 0.788 |
| | | $DICE$ - $l1_{max}$ | **0.744** | **0.758** | 0.761 | 0.762 | 0.774 | 0.777 | 0.789 | **0.794** | **0.795** |
| | | $ILS$ - $wasserstein_{min}$ | **0.744** | 0.751 | 0.757 | 0.759 | 0.772 | **0.789** | **0.794** | 0.792 | 0.791 |
| | xgboost | PlugInRule | **0.744** | 0.749 | 0.752 | 0.758 | 0.771 | 0.777 | 0.782 | 0.790 | 0.813 |
| | | PlugInRuleAUC | **0.744** | 0.749 | 0.752 | 0.764 | 0.765 | 0.763 | 0.761 | 0.772 | 0.794 |
| | | $ILS_{latent}$ - $l2^{\gamma}_{mean}$ | **0.744** | **0.750** | 0.755 | 0.760 | **0.777** | 0.788 | **0.803** | **0.808** | **0.814** |
| | | $ILS_{latent}$ - $inf^{\gamma}_{max}$ | **0.744** | 0.748 | 0.749 | 0.749 | 0.762 | **0.790** | 0.789 | 0.799 | 0.805 |
| | | $ILS_{latent}$ - $inf^{\gamma}_{mean}$ | **0.744** | 0.746 | 0.750 | 0.749 | 0.766 | 0.774 | 0.788 | 0.798 | 0.811 |
| | | $ILS_{latent}$ - $cosine^{\gamma}_{max}$ | **0.744** | 0.744 | **0.762** | **0.765** | 0.777 | 0.772 | 0.776 | 0.785 | 0.798 |
| | | $ILS_{latent}$ - $cosine^{\gamma}_{mean}$ | **0.744** | 0.744 | 0.760 | 0.762 | 0.767 | 0.781 | 0.779 | 0.778 | 0.804 |
| | LGBM | PlugInRule | **0.744** | 0.755 | **0.769** | **0.790** | **0.797** | **0.805** | 0.811 | 0.827 | **0.843** |
| | | PlugInRuleAUC | **0.744** | 0.744 | 0.753 | 0.753 | 0.752 | 0.757 | 0.756 | 0.761 | 0.780 |
| | | $LORE$ - $l1_{min}$ | **0.744** | 0.756 | 0.758 | 0.766 | 0.772 | 0.794 | **0.815** | 0.825 | 0.836 |
| | | $LORE$ - $braycurtis_{min}$ | **0.744** | 0.756 | 0.759 | 0.775 | 0.774 | 0.789 | 0.808 | 0.816 | 0.824 |
| | | $LORE$ - $l2_{min}$ | **0.744** | 0.756 | 0.757 | 0.762 | 0.772 | 0.787 | 0.806 | **0.827** | 0.838 |
| | | $LORE$ - $wasserstein^{\gamma}_{min}$ | **0.744** | 0.758 | 0.762 | 0.768 | 0.776 | 0.786 | 0.795 | 0.818 | 0.832 |
| | | $LORE$ - $wasserstein_{min}$ | **0.744** | **0.760** | 0.762 | 0.763 | 0.777 | 0.783 | 0.790 | 0.823 | 0.833 |
| Adult | RF | PlugInRule | **0.866** | 0.879 | 0.895 | 0.912 | 0.923 | 0.937 | 0.945 | 0.946 | 0.967 |
| | | PlugInRuleAUC | **0.866** | 0.872 | 0.868 | 0.875 | 0.875 | 0.883 | 0.881 | 0.894 | 0.913 |
| | | $ILS_{latent}$ - $cosine^{\gamma}_{min}$ | **0.866** | 0.866 | 0.889 | 0.925 | **0.954** | **0.964** | **0.963** | **0.971** | **0.976** |
| | | $ILS_{latent}$ - $cosine^{\gamma}_{mean}$ | **0.866** | **0.893** | **0.918** | 0.926 | 0.930 | 0.937 | 0.944 | 0.946 | 0.950 |
| | | $ILS$ - $braycurtis^{\gamma}_{min}$ | **0.866** | 0.870 | 0.907 | **0.929** | 0.939 | 0.942 | 0.949 | 0.950 | 0.958 |
| | | $LORE$ - $wasserstein^{\gamma}_{min}$ | **0.866** | 0.887 | 0.899 | 0.909 | 0.920 | 0.932 | 0.943 | 0.954 | 0.963 |
| | | $ILS_{latent}$ - $cosine_{mean}$ | **0.866** | 0.879 | 0.898 | 0.913 | 0.918 | 0.936 | 0.939 | 0.950 | 0.970 |
| | | $ILS_{latent}$ - $braycurtis_{min}$ | **0.866** | 0.876 | 0.893 | 0.910 | 0.925 | 0.937 | 0.947 | 0.947 | 0.964 |
| | mlp | PlugInRule | **0.864** | 0.878 | 0.890 | 0.905 | 0.914 | 0.927 | **0.941** | **0.950** | 0.967 |
| | | PlugInRuleAUC | **0.864** | 0.867 | 0.867 | 0.871 | 0.877 | 0.897 | 0.906 | 0.911 | 0.942 |
| | | $LORE$ - $wasserstein^{\gamma}_{min}$ | **0.864** | 0.883 | 0.904 | **0.912** | **0.920** | 0.926 | 0.929 | 0.945 | 0.958 |
| | | $ILS_{latent}$ - $cosine_{mean}$ | **0.864** | 0.878 | 0.902 | 0.910 | 0.911 | 0.927 | 0.934 | 0.944 | 0.952 |
| | | $ILS_{latent}$ - $braycurtis_{mean}$ | **0.864** | 0.877 | 0.895 | 0.905 | 0.913 | **0.929** | 0.932 | 0.940 | **0.967** |
| | | $ILS_{latent}$ - $cosine^{\gamma}_{mean}$ | **0.864** | **0.892** | **0.908** | 0.909 | 0.916 | 0.921 | 0.929 | 0.932 | 0.940 |
| | | $ILS_{latent}$ - $inf^{\gamma}_{min}$ | **0.864** | 0.889 | 0.907 | 0.909 | 0.919 | 0.923 | 0.926 | 0.932 | 0.946 |
| | xgboost | PlugInRule | **0.872** | 0.891 | 0.905 | 0.922 | 0.925 | 0.932 | 0.941 | 0.947 | **0.980** |
| | | PlugInRuleAUC | **0.872** | 0.870 | 0.876 | 0.876 | 0.887 | 0.892 | 0.898 | 0.916 | 0.935 |
| | | $ILS_{latent}$ - $cosine^{\gamma}_{mean}$ | **0.872** | **0.907** | 0.920 | 0.931 | 0.934 | 0.939 | 0.941 | 0.945 | 0.949 |
| | | $ILS_{latent}$ - $cosine^{\gamma}_{max}$ | **0.872** | 0.905 | 0.921 | **0.933** | 0.930 | 0.940 | 0.941 | 0.943 | 0.944 |
| | | $ILS_{latent}$ - $cosine^{\gamma}_{min}$ | **0.872** | 0.902 | 0.924 | 0.926 | 0.934 | 0.939 | 0.943 | 0.946 | 0.952 |
| | | $ILS_{latent}$ - $sqeuclidean^{\gamma}_{mean}$ | **0.872** | 0.898 | **0.925** | 0.930 | 0.934 | **0.942** | 0.943 | 0.942 | 0.945 |
| | | $ILS$ - $inf^{\gamma}_{mean}$ | **0.872** | 0.896 | 0.915 | 0.929 | **0.937** | 0.939 | **0.943** | **0.950** | 0.950 |
| | LGBM | PlugInRule | **0.870** | 0.889 | 0.903 | 0.920 | 0.924 | 0.937 | 0.941 | **0.953** | **0.970** |
| | | PlugInRuleAUC | **0.870** | 0.872 | 0.868 | 0.873 | 0.882 | 0.890 | 0.895 | 0.900 | 0.941 |
| | | $ILS_{latent}$ - $wasserstein^{\gamma}_{min}$ | **0.870** | 0.892 | 0.925 | **0.938** | 0.938 | **0.946** | **0.946** | 0.952 | 0.951 |
| | | $ILS_{latent}$ - $cosine^{\gamma}_{mean}$ | **0.870** | **0.907** | **0.926** | 0.933 | 0.934 | 0.938 | 0.939 | 0.941 | 0.950 |
| | | $ILS_{latent}$ - $inf^{\gamma}_{min}$ | **0.870** | 0.887 | 0.919 | 0.933 | **0.938** | 0.941 | 0.941 | 0.944 | 0.957 |
| | | $ILS_{latent}$ - $minkowski^{\gamma}_{mean}$ | **0.870** | 0.905 | 0.918 | 0.926 | 0.934 | 0.940 | 0.942 | 0.944 | 0.947 |
| | | $ILS_{latent}$ - $l2^{\gamma}_{mean}$ | **0.870** | 0.905 | 0.918 | 0.926 | 0.934 | 0.940 | 0.942 | 0.944 | 0.947 |

Table 6: Non-Rejected Accuracy of the top `SC-CE` combinations and Selective Classification baselines for the dataset `German` and `Adult`.

| Dataset | Black Box | Rejection Policy | Target Coverages | | | | | | | | |
|---|---|---|---|---|---|---|---|---|---|---|---|
| | | | 99% | 95% | 90% | 85% | 80% | 75% | 70% | 65% | 60% |
| Wisconsin | RF | PlugInRule | **0.965** | 0.985 | 0.984 | 0.992 | 0.991 | **1.000** | **1.000** | **1.000** | **1.000** |
| | | PlugInRuleAUC | **0.965** | 0.965 | 0.964 | 0.961 | 0.961 | 0.957 | 0.955 | 0.938 | 0.933 |
| | | DICE - $l1_{min}$ (D) | **0.965** | 0.992 | **1.000** | **1.000** | **1.000** | **1.000** | **1.000** | **1.000** | **1.000** |
| | | DICE - $l2_{mean}$ (D) | **0.965** | 0.992 | **1.000** | **1.000** | **1.000** | **1.000** | **1.000** | **1.000** | **1.000** |
| | | DICE - $braycurtis^{\gamma}_{min}$ | **0.965** | **0.993** | **1.000** | **1.000** | **1.000** | **1.000** | **1.000** | **1.000** | **1.000** |
| | | DICE - $braycurtis_{min}$ | **0.965** | 0.978 | **1.000** | **1.000** | **1.000** | **1.000** | **1.000** | **1.000** | **1.000** |
| | | DICE - $braycurtis^{\gamma}_{mean}$ | **0.965** | 0.985 | **1.000** | **1.000** | **1.000** | **1.000** | **1.000** | **1.000** | **1.000** |
| | mlp | PlugInRule | **0.979** | 0.979 | 0.985 | 0.984 | 0.984 | 0.990 | **1.000** | **1.000** | **1.000** |
| | | PlugInRuleAUC | **0.979** | 0.979 | 0.978 | 0.978 | 0.977 | 0.975 | 0.974 | 0.972 | 0.978 |
| | | LORE - $cosine^{\gamma}_{min}$ | **0.979** | 0.993 | 0.992 | **1.000** | **1.000** | **1.000** | **1.000** | **1.000** | **1.000** |
| | | LORE - $cosine_{min}$ | **0.979** | 0.986 | **0.992** | 0.992 | **1.000** | **1.000** | **1.000** | **1.000** | **1.000** |
| | | LORE - $braycurtis^{\gamma}_{min}$ | **0.979** | **0.993** | **0.992** | 0.992 | **1.000** | **1.000** | **1.000** | **1.000** | **1.000** |
| | | LORE - $braycurtis_{min}$ | **0.979** | 0.986 | **0.992** | 0.992 | **1.000** | **1.000** | **1.000** | **1.000** | **1.000** |
| | | LORE - $l2_{min}$ (D) | **0.979** | 0.986 | 0.985 | **1.000** | **1.000** | **1.000** | **1.000** | **1.000** | **1.000** |
| | xgboost | PlugInRule | **0.979** | **0.993** | 0.993 | 0.992 | 0.990 | **1.000** | **1.000** | **1.000** | **1.000** |
| | | PlugInRuleAUC | **0.979** | 0.978 | 0.978 | 0.977 | 0.977 | 0.977 | 0.977 | 0.975 | 0.971 |
| | | ILS - $cosine_{max}$ | **0.979** | 0.985 | **1.000** | **1.000** | **1.000** | **1.000** | **1.000** | **1.000** | **1.000** |
| | | $ILS_{latent}$ - $l2_{max}$ (D) | **0.979** | **0.993** | 0.992 | **1.000** | **1.000** | **1.000** | **1.000** | **1.000** | **1.000** |
| | | $ILS_{latent}$ - $l1_{max}$ (D) | **0.979** | **0.993** | 0.992 | **1.000** | **1.000** | **1.000** | **1.000** | **1.000** | **1.000** |
| | | $ILS_{latent}$ - $inf_{max}$ (D) | **0.979** | 0.986 | 0.992 | **1.000** | **1.000** | **1.000** | **1.000** | **1.000** | **1.000** |
| | | ILS - $cosine^{\gamma}_{max}$ | **0.979** | 0.985 | 0.992 | **1.000** | **1.000** | **1.000** | **1.000** | **1.000** | **1.000** |
| | LGBM | PlugInRule | **0.972** | **0.993** | 0.992 | 0.991 | 0.991 | 0.990 | 0.990 | 0.989 | **1.000** |
| | | PlugInRuleAUC | **0.972** | 0.970 | 0.969 | 0.968 | 0.967 | 0.975 | 0.973 | 0.969 | 0.966 |
| | | $ILS_{latent}$ - $cosine_{max}$ | **0.972** | **0.993** | **1.000** | **1.000** | **1.000** | **1.000** | **1.000** | **1.000** | **1.000** |
| | | ILS - $cosine^{\gamma}_{min}$ | **0.972** | 0.992 | 0.992 | 0.992 | 0.992 | **1.000** | **1.000** | **1.000** | **1.000** |
| | | $ILS_{latent}$ - $cosine^{\gamma}_{max}$ | **0.972** | **0.993** | 0.993 | 0.993 | 0.992 | **1.000** | **1.000** | **1.000** | **1.000** |
| | | $ILS_{latent}$ - $cosine_{mean}$ | **0.972** | **0.993** | 0.992 | 0.992 | **1.000** | **1.000** | **1.000** | **1.000** | **1.000** |
| | | DICE - $l1_{mean}$ (D) | **0.972** | 0.978 | 0.984 | **1.000** | **1.000** | **1.000** | **1.000** | **1.000** | **1.000** |
| Two-Moons | RF | PlugInRule | **0.916** | 0.933 | 0.951 | 0.960 | 0.965 | 0.973 | 0.978 | 0.988 | 0.993 |
| | | PlugInRuleAUC | **0.916** | 0.919 | 0.928 | 0.938 | 0.962 | 0.959 | 0.973 | 0.976 | 0.986 |
| | | DICE - $inf^{\gamma}_{max}$ | **0.916** | 0.937 | **0.968** | 0.980 | 0.980 | **0.989** | 0.988 | 0.994 | **1.000** |
| | | DICE - $wasserstein^{\gamma}_{max}$ | **0.916** | 0.936 | 0.963 | **0.985** | 0.985 | **0.989** | **0.989** | 0.988 | 0.993 |
| | | LORE - $inf_{mean}$ | **0.916** | **0.940** | 0.950 | 0.980 | **0.990** | **0.989** | 0.988 | **1.000** | **1.000** |
| | | DICE - $l2^{\gamma}_{max}$ | **0.916** | 0.937 | 0.963 | 0.980 | 0.985 | 0.989 | **0.989** | 0.994 | 0.993 |
| | | DICE - $l1^{\gamma}_{max}$ | **0.916** | 0.932 | 0.959 | **0.985** | 0.985 | **0.989** | **0.989** | 0.988 | 0.993 |
| | mlp | PlugInRule | **0.912** | **0.952** | 0.968 | 0.976 | 0.980 | **0.995** | 0.994 | 0.994 | **1.000** |
| | | PlugInRuleAUC | **0.912** | 0.911 | 0.956 | 0.968 | 0.980 | 0.990 | 0.989 | 0.994 | 0.993 |
| | | DICE - $l1^{\gamma}_{min}$ | **0.912** | 0.947 | 0.972 | **0.990** | 0.990 | 0.989 | 0.988 | 0.994 | **1.000** |
| | | LORE - $l1^{\gamma}_{min}$ | **0.912** | 0.943 | **0.977** | 0.985 | 0.985 | 0.989 | **0.994** | 0.994 | **1.000** |
| | | DICE - $sqeuclidean^{\gamma}_{min}$ | **0.912** | 0.943 | 0.972 | 0.986 | **0.990** | 0.989 | 0.989 | 0.994 | **1.000** |
| | | LORE - $l1_{min}$ | **0.912** | 0.943 | 0.972 | 0.977 | 0.985 | 0.989 | 0.994 | 0.994 | **1.000** |
| | | LORE - $l2_{min}$ | **0.912** | 0.944 | 0.968 | 0.977 | 0.986 | 0.984 | 0.988 | **1.000** | **1.000** |
| | xgboost | PlugInRule | **0.928** | 0.935 | 0.941 | 0.962 | 0.961 | 0.962 | 0.960 | 0.971 | 0.966 |
| | | PlugInRuleAUC | **0.928** | 0.937 | 0.950 | 0.946 | 0.955 | 0.954 | 0.964 | 0.981 | 0.980 |
| | | DICE - $l2_{max}$ | **0.928** | **0.958** | **0.982** | 0.982 | **0.990** | 0.989 | 0.994 | **1.000** | **1.000** |
| | | DICE - $inf_{max}$ | **0.928** | **0.958** | 0.978 | 0.986 | 0.986 | **0.989** | 0.994 | **1.000** | **1.000** |
| | | DICE - $inf^{\gamma}_{max}$ | **0.928** | 0.934 | **0.982** | 0.986 | 0.990 | 0.989 | **1.000** | **1.000** | **1.000** |
| | | DICE - $l2^{\gamma}_{max}$ | **0.928** | 0.935 | **0.982** | **0.990** | 0.990 | 0.989 | 0.994 | **1.000** | **1.000** |
| | | DICE - $sqeuclidean^{\gamma}_{max}$ | **0.928** | 0.939 | 0.973 | 0.986 | 0.990 | 0.989 | 0.994 | **1.000** | **1.000** |
| | LGBM | PlugInRule | **0.936** | 0.944 | 0.953 | 0.959 | 0.960 | 0.973 | 0.972 | 0.970 | 0.985 |
| | | PlugInRuleAUC | **0.936** | 0.937 | 0.936 | 0.943 | 0.945 | 0.967 | 0.966 | 0.974 | 0.986 |
| | | DICE - $wasserstein^{\gamma}_{mean}$ | **0.936** | 0.946 | 0.964 | **0.981** | **0.995** | 0.995 | **0.994** | 0.994 | **1.000** |
| | | DICE - $l2^{\gamma}_{max}$ | **0.936** | **0.950** | 0.969 | 0.976 | 0.985 | 0.995 | 0.994 | 0.994 | **1.000** |
| | | DICE - $wasserstein^{\gamma}_{max}$ | **0.936** | 0.950 | **0.969** | 0.976 | 0.990 | 0.990 | **0.994** | 0.994 | **1.000** |
| | | DICE - $inf^{\gamma}_{max}$ | **0.936** | **0.950** | 0.969 | 0.977 | 0.980 | 0.995 | 0.994 | **0.994** | **1.000** |
| | | DICE - $l2^{\gamma}_{mean}$ | **0.936** | 0.942 | 0.960 | **0.981** | 0.990 | **0.995** | 0.994 | 0.994 | **1.000** |

Table 7: Non-Rejected Accuracy of the top L2R-CE combinations and Selective Classification baselines for the datasets Wisconsin and Two-Moons.

| Dataset | Black Box | Rejection Policy | Target Coverages | | | | | | | | |
|---|---|---|---|---|---|---|---|---|---|---|---|
| | | | 99% | 95% | 90% | 85% | 80% | 75% | 70% | 65% | 60% |
| German | RF | PlugInRule | **0.752** | 0.744 | 0.744 | **0.764** | **0.764** | 0.740 | 0.740 | 0.732 | 0.720 |
| | | PlugInRuleAUC | **0.752** | 0.744 | 0.744 | 0.748 | 0.720 | 0.712 | 0.680 | 0.648 | 0.620 |
| | | ILS - $\mathrm{mae}^{\gamma}_{max}$ | **0.752** | 0.760 | 0.760 | 0.760 | 0.760 | **0.760** | **0.760** | 0.764 | **0.768** |
| | | ILS - $\mathrm{l1}^{\gamma}_{max}$ | **0.752** | 0.760 | 0.760 | 0.760 | 0.760 | **0.760** | **0.760** | 0.760 | 0.760 |
| | | ILS - $\mathrm{l1}_{mean}$ | **0.752** | 0.764 | **0.764** | 0.752 | 0.760 | **0.760** | 0.756 | 0.760 | 0.764 |
| | | ILS - $\mathrm{sqeuclidean}_{mean}$ | **0.752** | **0.776** | 0.752 | **0.764** | 0.752 | 0.752 | **0.760** | 0.756 | 0.752 |
| | | ILS - $\mathrm{wasserstein}^{\gamma}_{max}$ | **0.752** | 0.756 | 0.756 | 0.756 | 0.756 | 0.756 | 0.756 | 0.756 | 0.760 |
| | mlp | PlugInRule | **0.744** | 0.748 | 0.712 | 0.700 | 0.692 | 0.684 | 0.664 | 0.660 | 0.640 |
| | | PlugInRuleAUC | **0.744** | 0.716 | 0.712 | 0.684 | 0.680 | 0.672 | 0.660 | 0.628 | 0.596 |
| | | LORE - $\mathrm{wasserstein}^{\gamma}_{max}$ | **0.744** | 0.744 | 0.744 | 0.744 | 0.744 | **0.748** | 0.728 | **0.736** | 0.652 |
| | | DICE - $\mathrm{cosine}^{\gamma}_{max}$ | **0.744** | **0.752** | 0.764 | **0.756** | 0.744 | 0.736 | 0.728 | 0.696 | 0.660 |
| | | LORE - $\mathrm{mae}^{\gamma}_{mean}$ | **0.744** | 0.748 | 0.744 | 0.744 | 0.740 | 0.732 | **0.736** | 0.724 | 0.652 |
| | | DICE - $\mathrm{braycurtis}_{max}$ | **0.744** | 0.748 | 0.740 | 0.744 | **0.752** | 0.732 | 0.728 | 0.712 | **0.680** |
| | | DICE - $\mathrm{cosine}_{max}$ | **0.744** | **0.752** | **0.768** | 0.744 | 0.744 | 0.736 | 0.728 | 0.696 | 0.668 |
| | xgboost | PlugInRule | **0.744** | 0.740 | 0.736 | 0.724 | 0.728 | 0.724 | 0.720 | 0.720 | 0.712 |
| | | PlugInRuleAUC | **0.744** | 0.740 | 0.712 | 0.720 | 0.716 | 0.700 | 0.676 | 0.668 | 0.656 |
| | | ILS - $\mathrm{l2}^{\gamma}_{max}$ | **0.744** | **0.748** | **0.748** | **0.748** | **0.748** | **0.748** | 0.748 | 0.748 | 0.748 |
| | | ILS - $\mathrm{inf}_{min}$ | **0.744** | **0.748** | **0.748** | **0.748** | **0.748** | **0.748** | 0.748 | 0.748 | 0.748 |
| | | ILS - $\mathrm{inf}^{\gamma}_{mean}$ | **0.744** | 0.740 | 0.740 | 0.740 | 0.740 | 0.740 | **0.752** | **0.752** | **0.752** |
| | | ILS - $\mathrm{mae}^{\gamma}_{min}$ | **0.744** | 0.744 | 0.744 | 0.744 | 0.744 | 0.744 | 0.744 | 0.744 | 0.724 |
| | | ILS - $\mathrm{l1}^{\gamma}_{max}$ | **0.744** | 0.740 | 0.740 | 0.740 | 0.740 | 0.740 | 0.744 | 0.744 | 0.744 |
| | | $\mathrm{ILS}_{latent}$ - $\mathrm{l2}^{\gamma}_{min}$ | **0.744** | 0.744 | 0.744 | 0.744 | 0.744 | 0.744 | 0.740 | 0.736 | 0.720 |
| | LGBM | PlugInRule | **0.744** | **0.756** | **0.768** | **0.776** | **0.772** | **0.768** | **0.768** | **0.768** | 0.744 |
| | | PlugInRuleAUC | **0.744** | 0.728 | 0.732 | 0.716 | 0.696 | 0.680 | 0.660 | 0.648 | 0.644 |
| | | LORE - $\mathrm{sqeuclidean}^{\gamma}_{min}$ | **0.744** | 0.748 | 0.748 | 0.748 | 0.748 | 0.748 | 0.748 | 0.748 | 0.748 |
| | | LORE - $\mathrm{mae}^{\gamma}_{min}$ | **0.744** | 0.752 | 0.748 | 0.752 | 0.764 | 0.744 | 0.744 | 0.760 | **0.760** |
| | | LORE - $\mathrm{braycurtis}_{min}$ | **0.744** | 0.752 | 0.756 | 0.764 | 0.752 | 0.748 | 0.756 | 0.752 | 0.712 |
| | | LORE - $\mathrm{l1}^{\gamma}_{min}$ | **0.744** | 0.748 | 0.752 | 0.752 | 0.748 | 0.748 | 0.740 | 0.760 | 0.756 |
| | | LORE - $\mathrm{l1}_{min}$ | **0.744** | 0.752 | 0.752 | 0.756 | 0.744 | 0.748 | 0.760 | 0.760 | 0.716 |
| Adult | RF | PlugInRule | **0.866** | 0.860 | 0.842 | 0.826 | 0.816 | 0.796 | 0.786 | 0.756 | 0.700 |
| | | PlugInRuleAUC | **0.866** | 0.846 | 0.806 | 0.792 | 0.770 | 0.748 | 0.712 | 0.698 | 0.666 |
| | | DICE - $\mathrm{l2}^{\gamma}_{min}$ | **0.866** | **0.872** | **0.872** | **0.872** | **0.872** | **0.872** | 0.872 | 0.872 | 0.872 |
| | | DICE - $\mathrm{mae}^{\gamma}_{min}$ | **0.866** | **0.872** | **0.872** | **0.872** | **0.872** | **0.872** | 0.872 | 0.874 | 0.866 |
| | | DICE - $\mathrm{mae}^{\gamma}_{max}$ | **0.866** | 0.868 | 0.868 | 0.868 | 0.868 | 0.870 | **0.874** | **0.876** | 0.872 |
| | | LORE - $\mathrm{inf}^{\gamma}_{min}$ | **0.866** | 0.870 | 0.870 | 0.870 | 0.870 | 0.870 | 0.870 | 0.870 | 0.870 |
| | | DICE - $\mathrm{wasserstein}^{\gamma}_{max}$ | **0.866** | 0.868 | 0.868 | 0.868 | 0.868 | 0.868 | 0.870 | 0.874 | **0.876** |
| | mlp | PlugInRule | **0.864** | 0.852 | 0.856 | 0.852 | 0.830 | 0.818 | 0.792 | 0.750 | 0.694 |
| | | PlugInRuleAUC | **0.864** | 0.808 | 0.784 | 0.772 | 0.758 | 0.738 | 0.720 | 0.708 | 0.680 |
| | | DICE - $\mathrm{inf}^{\gamma}_{max}$ | **0.864** | **0.870** | **0.870** | **0.870** | **0.870** | **0.870** | **0.870** | **0.870** | **0.870** |
| | | DICE - $\mathrm{inf}^{\gamma}_{mean}$ | **0.864** | **0.870** | **0.870** | **0.870** | **0.870** | **0.870** | **0.870** | **0.870** | 0.868 |
| | | LORE - $\mathrm{sqeuclidean}^{\gamma}_{min}$ | **0.864** | 0.868 | 0.868 | 0.868 | 0.868 | 0.868 | 0.868 | 0.868 | 0.868 |
| | | DICE - $\mathrm{inf}^{\gamma}_{min}$ | **0.864** | 0.868 | 0.868 | 0.868 | 0.868 | 0.868 | 0.868 | 0.868 | 0.866 |
| | | DICE - $\mathrm{mae}^{\gamma}_{min}$ | **0.864** | 0.866 | 0.866 | 0.866 | 0.868 | 0.868 | 0.868 | 0.868 | 0.868 |
| | xgboost | PlugInRule | **0.872** | **0.876** | 0.864 | 0.862 | 0.836 | 0.818 | 0.786 | 0.738 | 0.702 |
| | | PlugInRuleAUC | **0.872** | 0.834 | 0.808 | 0.782 | 0.772 | 0.752 | 0.724 | 0.720 | 0.686 |
| | | ILS - $\mathrm{sqeuclidean}^{\gamma}_{max}$ | **0.872** | **0.876** | **0.876** | **0.876** | **0.876** | **0.876** | **0.876** | **0.876** | **0.876** |
| | | DICE - $\mathrm{wasserstein}^{\gamma}_{mean}$ | **0.872** | 0.874 | 0.874 | 0.874 | 0.874 | 0.874 | 0.874 | 0.874 | 0.874 |
| | | DICE - $\mathrm{inf}^{\gamma}_{min}$ | **0.872** | 0.872 | 0.872 | 0.872 | 0.872 | 0.872 | 0.872 | 0.872 | 0.872 |
| | | ILS - $\mathrm{sqeuclidean}^{\gamma}_{min}$ | **0.872** | 0.872 | 0.872 | 0.872 | 0.872 | 0.872 | 0.872 | 0.872 | 0.870 |
| | | DICE - $\mathrm{l1}^{\gamma}_{max}$ | **0.872** | 0.870 | 0.870 | 0.870 | 0.870 | 0.870 | 0.870 | 0.870 | 0.870 |
| | LGBM | PlugInRule | **0.870** | 0.872 | 0.864 | 0.860 | 0.846 | 0.826 | 0.788 | 0.742 | 0.702 |
| | | PlugInRuleAUC | **0.870** | 0.830 | 0.796 | 0.778 | 0.766 | 0.754 | 0.732 | 0.720 | 0.702 |
| | | DICE - $\mathrm{inf}^{\gamma}_{max}$ | **0.870** | **0.874** | **0.874** | **0.874** | **0.874** | **0.874** | **0.874** | **0.874** | 0.874 |
| | | DICE - $\mathrm{mae}^{\gamma}_{max}$ | **0.870** | 0.872 | 0.872 | 0.872 | 0.872 | 0.872 | 0.872 | 0.872 | **0.880** |
| | | DICE - $\mathrm{wasserstein}^{\gamma}_{mean}$ | **0.870** | 0.872 | 0.872 | 0.872 | 0.872 | 0.872 | 0.872 | **0.874** | 0.876 |
| | | ILS - $\mathrm{wasserstein}^{\gamma}_{max}$ | **0.870** | 0.872 | 0.872 | 0.872 | 0.872 | **0.874** | **0.874** | 0.872 | 0.872 |
| | | DICE - $\mathrm{wasserstein}^{\gamma}_{max}$ | **0.870** | 0.872 | 0.872 | 0.872 | 0.872 | 0.872 | 0.872 | 0.872 | 0.872 |

Table 8: Classification Quality of the top `SC-CE` combinations and Selective Classification baselines for the dataset `German` and `Adult`.

| Dataset | Black Box | Rejection Policy | Target Coverages | | | | | | | | |
|---|---|---|---|---|---|---|---|---|---|---|---|
| | | | 99% | 95% | 90% | 85% | 80% | 75% | 70% | 65% | 60% |
| Wisconsin | RF | PlugInRule | **0.965** | 0.944 | 0.887 | 0.880 | 0.810 | 0.746 | 0.718 | 0.690 | 0.585 |
| | | PlugInRuleAUC | **0.965** | 0.958 | 0.937 | 0.859 | 0.859 | 0.782 | 0.754 | 0.535 | 0.493 |
| | | LORE - $l1^{\gamma}_{min}$ | **0.965** | 0.965 | 0.965 | **0.965** | 0.965 | 0.965 | **0.965** | 0.965 | **0.965** |
| | | LORE - $sqeuclidean^{\gamma}_{min}$ | **0.965** | 0.965 | 0.965 | **0.965** | 0.965 | 0.965 | **0.965** | 0.972 | 0.958 |
| | | LORE - $inf^{\gamma}_{max}$ (D) | **0.965** | 0.958 | 0.958 | 0.958 | 0.958 | 0.958 | 0.958 | 0.958 | 0.958 |
| | | $ILS_{latent}$ - $wasserstein^{\gamma}_{min}$ | **0.965** | 0.965 | 0.965 | **0.965** | **0.979** | **0.979** | 0.965 | 0.930 | 0.894 |
| | | ILS - $sqeuclidean^{\gamma}_{max}$ | **0.965** | **0.979** | **0.972** | **0.965** | 0.951 | 0.944 | 0.944 | 0.944 | 0.944 |
| | mlp | PlugInRule | **0.979** | 0.972 | 0.930 | 0.894 | 0.859 | 0.746 | 0.711 | 0.690 | 0.577 |
| | | PlugInRuleAUC | **0.979** | 0.972 | 0.951 | 0.923 | 0.880 | 0.810 | 0.796 | 0.739 | 0.641 |
| | | $ILS_{latent}$ - $l1^{\gamma}_{max}$ | **0.979** | **0.979** | **0.979** | **0.979** | **0.979** | 0.972 | **0.972** | **0.972** | 0.908 |
| | | $ILS_{latent}$ - $l2^{\gamma}_{max}$ | **0.979** | **0.979** | **0.979** | **0.979** | **0.979** | 0.972 | **0.972** | **0.972** | 0.915 |
| | | $ILS_{latent}$ - $inf^{\gamma}_{max}$ (D) | **0.979** | 0.972 | 0.972 | 0.972 | 0.958 | 0.951 | 0.944 | 0.915 | 0.796 |
| | | LORE - $mae^{\gamma}_{min}$ | **0.979** | **0.979** | **0.979** | **0.979** | **0.979** | **0.979** | **0.972** | 0.944 | 0.500 |
| | | $ILS_{latent}$ - $inf^{\gamma}_{mean}$ (D) | **0.979** | 0.972 | 0.972 | 0.965 | 0.944 | 0.923 | 0.894 | 0.859 | 0.690 |
| | xgboost | PlugInRule | **0.979** | 0.972 | 0.951 | 0.859 | 0.711 | 0.648 | 0.620 | 0.577 | 0.514 |
| | | PlugInRuleAUC | **0.979** | 0.958 | 0.937 | 0.915 | 0.915 | 0.894 | 0.880 | 0.824 | 0.711 |
| | | ILS - $sqeuclidean^{\gamma}_{max}$ | **0.979** | **0.979** | **0.979** | **0.979** | **0.979** | **0.979** | **0.979** | **0.979** | **0.979** |
| | | $ILS_{latent}$ - $braycurtis^{\gamma}_{max}$ | **0.979** | **0.979** | 0.972 | 0.965 | 0.965 | 0.965 | 0.965 | 0.965 | 0.930 |
| | | LORE - $mae^{\gamma}_{min}$ | **0.979** | **0.979** | **0.979** | **0.979** | **0.979** | 0.965 | 0.958 | 0.951 | 0.859 |
| | | LORE - $inf^{\gamma}_{max}$ (D) | **0.979** | **0.979** | **0.979** | **0.979** | **0.979** | 0.965 | 0.958 | 0.930 | 0.824 |
| | | $ILS_{latent}$ - $cosine^{\gamma}_{max}$ | **0.979** | 0.958 | 0.972 | 0.972 | 0.972 | 0.958 | 0.937 | 0.930 | 0.866 |
| | LGBM | PlugInRule | **0.972** | **0.979** | 0.852 | 0.831 | 0.768 | 0.718 | 0.711 | 0.648 | 0.563 |
| | | PlugInRuleAUC | **0.972** | 0.908 | 0.866 | 0.852 | 0.831 | 0.817 | 0.768 | 0.662 | 0.606 |
| | | ILS - $mae^{\gamma}_{max}$ | **0.972** | 0.972 | **0.972** | 0.972 | **0.972** | **0.979** | **0.979** | **0.979** | 0.972 |
| | | $ILS_{latent}$ - $sqeuclidean^{\gamma}_{mean}$ | **0.972** | 0.972 | **0.972** | 0.972 | **0.972** | 0.972 | **0.979** | **0.979** | 0.972 |
| | | ILS - $sqeuclidean^{\gamma}_{mean}$ | **0.972** | 0.972 | **0.972** | 0.972 | **0.972** | 0.972 | 0.972 | 0.972 | **0.979** |
| | | $ILS_{latent}$ - $inf^{\gamma}_{max}$ (D) | **0.972** | 0.972 | **0.972** | 0.972 | **0.972** | 0.972 | 0.972 | 0.972 | 0.972 |
| | | ILS - $wasserstein^{\gamma}_{max}$ | **0.972** | 0.972 | **0.972** | **0.979** | 0.972 | **0.979** | **0.979** | 0.972 | 0.958 |
| Two-Moons | RF | PlugInRule | **0.916** | 0.916 | 0.888 | 0.828 | 0.828 | 0.796 | 0.792 | 0.740 | 0.660 |
| | | PlugInRuleAUC | **0.916** | 0.912 | 0.888 | 0.872 | 0.852 | 0.800 | 0.784 | 0.732 | 0.624 |
| | | LORE - $cosine^{\gamma}_{max}$ | **0.916** | **0.920** | **0.920** | **0.920** | **0.920** | **0.920** | **0.920** | **0.920** | **0.924** |
| | | $ILS_{latent}$ - $inf^{\gamma}_{min}$ | **0.916** | **0.920** | **0.920** | **0.920** | **0.920** | **0.920** | **0.920** | **0.920** | 0.920 |
| | | $ILS_{latent}$ - $l1^{\gamma}_{min}$ | **0.916** | **0.920** | **0.920** | **0.920** | **0.920** | **0.920** | 0.916 | 0.916 | 0.916 |
| | | $ILS_{latent}$ - $l2^{\gamma}_{min}$ | **0.916** | **0.920** | **0.920** | **0.920** | **0.920** | 0.916 | **0.920** | **0.920** | 0.916 |
| | | $ILS_{latent}$ - $wasserstein^{\gamma}_{min}$ | **0.916** | **0.920** | **0.920** | 0.916 | 0.916 | **0.920** | 0.916 | 0.912 | 0.908 |
| | mlp | PlugInRule | **0.912** | **0.912** | 0.904 | 0.892 | 0.872 | 0.812 | 0.780 | 0.744 | 0.656 |
| | | PlugInRuleAUC | **0.912** | 0.900 | 0.908 | 0.900 | 0.864 | 0.844 | 0.780 | 0.712 | 0.620 |
| | | LORE - $braycurtis^{\gamma}_{max}$ | **0.912** | **0.912** | **0.912** | 0.912 | 0.912 | **0.912** | 0.916 | 0.912 | **0.904** |
| | | ILS - $sqeuclidean^{\gamma}_{min}$ | **0.912** | 0.904 | 0.904 | 0.904 | 0.904 | 0.904 | 0.904 | 0.904 | **0.904** |
| | | LORE - $sqeuclidean^{\gamma}_{mean}$ | **0.912** | 0.908 | **0.912** | **0.920** | **0.920** | 0.904 | 0.888 | 0.888 | 0.848 |
| | | ILS - $braycurtis^{\gamma}_{min}$ | **0.912** | 0.900 | 0.900 | 0.900 | 0.900 | 0.900 | 0.900 | 0.900 | 0.896 |
| | | LORE - $cosine^{\gamma}_{max}$ | **0.912** | 0.892 | 0.888 | 0.884 | 0.876 | 0.860 | 0.856 | 0.852 | 0.796 |
| | xgboost | PlugInRule | **0.928** | **0.928** | 0.912 | 0.856 | 0.824 | 0.752 | 0.712 | 0.716 | 0.620 |
| | | PlugInRuleAUC | **0.928** | 0.908 | 0.868 | 0.800 | 0.796 | 0.784 | 0.688 | 0.692 | 0.636 |
| | | LORE - $braycurtis^{\gamma}_{max}$ | **0.928** | **0.928** | **0.928** | **0.928** | **0.928** | **0.928** | **0.928** | **0.928** | **0.920** |
| | | LORE - $braycurtis_{max}$ | **0.928** | 0.920 | 0.876 | 0.876 | 0.876 | 0.876 | 0.876 | 0.876 | 0.876 |
| | | LORE - $cosine^{\gamma}_{max}$ | **0.928** | 0.888 | 0.880 | 0.872 | 0.872 | 0.868 | 0.856 | 0.848 | 0.836 |
| | | ILS - $l1^{\gamma}_{min}$ | **0.928** | 0.924 | 0.904 | 0.884 | 0.860 | 0.836 | 0.816 | 0.808 | 0.808 |
| | | ILS - $wasserstein^{\gamma}_{min}$ | **0.928** | 0.924 | 0.908 | 0.884 | 0.844 | 0.824 | 0.808 | 0.808 | 0.764 |
| | LGBM | PlugInRule | **0.936** | **0.944** | 0.912 | 0.868 | 0.792 | 0.772 | 0.736 | 0.692 | 0.580 |
| | | PlugInRuleAUC | **0.936** | 0.840 | 0.828 | 0.808 | 0.772 | 0.736 | 0.712 | 0.640 | 0.600 |
| | | LORE - $cosine^{\gamma}_{max}$ | **0.936** | **0.944** | **0.932** | **0.932** | **0.908** | 0.888 | 0.864 | 0.852 | 0.840 |
| | | LORE - $braycurtis^{\gamma}_{max}$ | **0.936** | 0.916 | 0.900 | 0.900 | 0.896 | **0.892** | **0.888** | **0.888** | **0.888** |
| | | LORE - $braycurtis_{max}$ | **0.936** | 0.924 | 0.884 | 0.884 | 0.884 | 0.884 | 0.884 | 0.884 | 0.884 |
| | | DICE - $sqeuclidean^{\gamma}_{mean}$ | **0.936** | 0.932 | 0.920 | 0.888 | 0.876 | 0.852 | 0.816 | 0.768 | 0.672 |
| | | DICE - $l1_{max}$ | **0.936** | 0.920 | 0.916 | 0.908 | 0.880 | 0.848 | 0.816 | 0.780 | 0.668 |

Table 9: Classification Quality of the top SC-CE combinations and Selective Classification baselines for the dataset Wisconsin and Two-Moons.

| Dataset | Black Box | Rejection Policy | Target Coverages | | | | | | | | |
|---|---|---|---|---|---|---|---|---|---|---|---|
| | | | 99% | 95% | 90% | 85% | 80% | 75% | 70% | 65% | 60% |
| German | RF | PlugInRule | 0.000 | 1.516 | 1.516 | **4.169** | **3.639** | 2.637 | 2.637 | 2.471 | 2.274 |
| | | PlugInRuleAUC | 0.000 | 2.628 | 2.628 | 2.843 | 2.022 | 1.987 | 1.633 | 1.423 | 1.539 |
| | | ILS - $l2_{max}^{\gamma}$ | 0.000 | **21.226** | 4.043 | 3.249 | 2.713 | **3.032** | **3.032** | **3.032** | 2.426 |
| | | $ILS_{latent}$ - $sqeuclidean_{mean}^{\gamma}$ | 0.000 | 9.097 | **4.765** | 3.266 | 3.211 | 2.637 | 2.583 | 2.325 | 2.541 |
| | | $ILS_{latent}$ - $cosine_{max}$ | 0.000 | 15.161 | 3.032 | 3.032 | 3.222 | 2.888 | 2.491 | 2.445 | 2.622 |
| | | $ILS_{latent}$ - $sqeuclidean_{max}^{\gamma}$ | 0.000 | 15.161 | 2.780 | 3.465 | 2.599 | 2.491 | 2.543 | 2.665 | 2.729 |
| | | $ILS_{latent}$ - $l1_{max}^{\gamma}$ | 0.000 | 15.161 | 3.032 | 3.682 | 2.599 | 2.382 | 2.628 | 2.748 | **2.864** |
| | mlp | PlugInRule | 0.000 | 3.633 | 1.539 | 1.453 | 1.453 | 1.500 | 1.335 | 1.300 | 1.332 |
| | | PlugInRuleAUC | 0.000 | 0.872 | 1.356 | 1.162 | 1.246 | 1.321 | 1.341 | 1.186 | 1.277 |
| | | DICE - $cosine_{max}^{\gamma}$ | 0.000 | **5.812** | 6.539 | 3.699 | 2.906 | **2.629** | **2.441** | 1.849 | 1.635 |
| | | DICE - $cosine_{max}$ | 0.000 | 5.812 | 6.394 | 2.906 | 2.906 | **2.629** | 2.422 | 1.849 | **1.706** |
| | | DICE - $cosine_{mean}^{\gamma}$ | 0.000 | 5.812 | 4.069 | **4.359** | 2.393 | 2.616 | 2.105 | 1.776 | 1.643 |
| | | DICE - $cosine_{mean}$ | 0.000 | 5.812 | 3.321 | 2.906 | 2.735 | 2.491 | 2.105 | 1.776 | 1.679 |
| | | DICE - $braycurtis_{max}$ | 0.000 | 4.359 | 2.543 | 2.906 | **3.353** | 2.447 | 2.325 | **1.976** | 1.683 |
| | xgboost | PlugInRule | 0.000 | 2.180 | 2.076 | 1.868 | 2.260 | 2.246 | 2.209 | 2.283 | 2.294 |
| | | PlugInRuleAUC | 0.000 | 2.180 | 1.453 | 1.937 | 1.889 | 1.627 | 1.409 | 1.526 | 1.653 |
| | | ILS - $l2_{max}^{\gamma}$ | 0.000 | 5.812 | 5.812 | 5.812 | 5.812 | **5.812** | **5.812** | **5.812** | **5.812** |
| | | ILS - $inf_{max}^{\gamma}$ | 0.000 | **8.719** | **8.719** | **8.719** | 1.661 | 1.332 | 1.176 | 1.097 | 1.279 |
| | | ILS - $sqeuclidean_{max}^{\gamma}$ | 0.000 | 5.812 | 5.812 | 5.812 | **8.719** | 1.661 | 1.372 | 1.182 | 1.198 |
| | | ILS - $inf_{min}$ | 0.000 | 5.812 | 3.633 | 3.633 | 3.633 | 3.633 | 3.633 | 3.633 | 3.633 |
| | | ILS - $inf_{min}^{\gamma}$ | 0.000 | **8.719** | 2.180 | 2.180 | 1.310 | 1.274 | 1.274 | 1.177 | 1.235 |
| | | ILS - $mae_{max}^{\gamma}$ | 0.000 | 0.000 | 2.906 | 2.906 | 5.812 | **5.812** | 1.453 | 1.643 | 1.258 |
| | LGBM | PlugInRule | 0.000 | **11.625** | **8.719** | 5.490 | 4.471 | 3.932 | 3.824 | 3.633 | 2.906 |
| | | PlugInRuleAUC | 0.000 | 0.969 | 1.937 | 1.550 | 1.321 | 1.356 | 1.257 | 1.284 | 1.482 |
| | | LORE - $sqeuclidean_{min}^{\gamma}$ | 0.000 | 5.812 | 5.812 | **5.812** | **5.812** | 5.812 | **5.812** | **5.812** | **3.633** |
| | | LORE - $braycurtis_{min}^{\gamma}$ | 0.000 | 2.906 | 2.906 | 2.906 | **5.812** | 14.531 | 4.844 | 4.359 | 2.806 |
| | | LORE - $mae_{min}^{\gamma}$ | 0.000 | **8.719** | 3.633 | 4.359 | 5.328 | 2.906 | 2.906 | 3.460 | 3.353 |
| | | LORE - $braycurtis_{min}$ | 0.000 | 4.844 | 5.812 | 4.982 | 3.435 | 3.068 | 3.303 | 3.130 | 2.339 |
| | | LORE - $l1_{min}^{\gamma}$ | 0.000 | 5.812 | 4.844 | 4.359 | 3.391 | 3.170 | 2.753 | 3.460 | 3.218 |
| Adult | RF | PlugInRule | 0.000 | 4.847 | 4.039 | 3.878 | 3.814 | 3.563 | 3.491 | 3.012 | 2.631 |
| | | PlugInRuleAUC | 0.000 | 2.424 | 1.223 | 1.583 | 1.459 | 1.636 | 1.436 | 1.659 | 1.813 |
| | | DICE - $l2_{min}^{\gamma}$ | 0.000 | **25.851** | **25.851** | **25.851** | **25.851** | **25.851** | 25.851 | 25.851 | **25.851** |
| | | DICE - $mae_{min}^{\gamma}$ | 0.000 | **25.851** | **25.851** | **25.851** | **25.851** | **25.851** | 25.851 | 19.388 | 6.463 |
| | | DICE - $inf_{min}^{\gamma}$ | 0.000 | 19.388 | 19.388 | 19.388 | 19.388 | 19.388 | 19.388 | 19.388 | 19.388 |
| | | DICE - $wasserstein_{max}^{\gamma}$ | 0.000 | 12.925 | 12.925 | 12.925 | 12.925 | 12.925 | 19.388 | 32.313 | 17.234 |
| | | DICE - $mae_{max}^{\gamma}$ | 0.000 | 0.000 | 0.000 | 12.925 | 12.925 | 19.388 | **32.313** | **38.776** | 9.694 |
| | mlp | PlugInRule | 0.000 | 3.971 | 5.143 | 5.162 | 4.149 | 3.996 | 3.564 | 3.000 | 2.577 |
| | | PlugInRuleAUC | 0.000 | 1.271 | 1.167 | 1.400 | 1.543 | 2.003 | 2.038 | 2.044 | 2.237 |
| | | DICE - $mae_{min}^{\gamma}$ | 0.000 | **12.706** | **12.706** | **12.706** | 19.059 | 19.059 | 19.059 | **19.059** | **19.059** |
| | | DICE - $wasserstein_{min}^{\gamma}$ | 0.000 | 0.000 | **12.706** | **12.706** | 12.706 | 12.706 | 19.059 | **19.059** | **19.059** |
| | | LORE - $sqeuclidean_{min}^{\gamma}$ | 0.000 | **12.706** | **12.706** | **12.706** | 12.706 | 12.706 | 12.706 | 12.706 | 12.706 |
| | | DICE - $l1_{max}^{\gamma}$ | 0.000 | 6.353 | 6.353 | **12.706** | 25.412 | 44.471 | 10.588 | 4.941 | 2.795 |
| | | DICE - $l1_{mean}^{\gamma}$ | 0.000 | 6.353 | 6.353 | **12.706** | 12.706 | 19.059 | 38.118 | 10.165 | 3.315 |
| | xgboost | PlugInRule | 0.000 | 8.175 | 5.722 | 5.839 | 4.408 | 3.938 | 3.366 | 2.773 | 2.763 |
| | | PlugInRuleAUC | 0.000 | 0.649 | 1.363 | 1.239 | 1.729 | 1.766 | 1.771 | 2.106 | 2.154 |
| | | ILS - $sqeuclidean_{mean}^{\gamma}$ | 0.000 | **20.438** | **20.438** | **20.438** | **20.438** | **20.438** | **20.438** | **20.438** | **20.438** |
| | | ILS - $sqeuclidean_{max}^{\gamma}$ | 0.000 | 13.625 | 13.625 | 13.625 | 13.625 | 13.625 | 13.625 | 13.625 | 13.625 |
| | | ILS - $braycurtis_{mean}^{\gamma}$ | 0.000 | 13.625 | **20.438** | 6.812 | 6.812 | 6.812 | 5.551 | 5.378 | 3.331 |
| | | ILS - $wasserstein_{max}^{\gamma}$ | 0.000 | 8.326 | 5.299 | 4.866 | 4.061 | 3.185 | 3.370 | 2.805 | 2.348 |
| | | ILS - $l1_{mean}^{\gamma}$ | 0.000 | 6.358 | 5.904 | 4.258 | 4.258 | 3.742 | 3.572 | 3.368 | 3.085 |
| | LGBM | PlugInRule | 0.000 | 7.301 | 5.856 | 5.736 | 4.908 | 4.318 | 3.426 | 2.902 | 2.677 |
| | | PlugInRuleAUC | 0.000 | 1.338 | 0.797 | 1.195 | 1.575 | 1.779 | 1.780 | 1.819 | 2.368 |
| | | DICE - $inf_{max}^{\gamma}$ | 0.000 | **20.077** | **20.077** | **20.077** | **20.077** | **20.077** | **20.077** | **20.077** | 20.077 |
| | | DICE - $mae_{max}^{\gamma}$ | 0.000 | 13.385 | 13.385 | 13.385 | 13.385 | 13.385 | 13.385 | 13.385 | **40.154** |
| | | DICE - $wasserstein_{mean}^{\gamma}$ | 0.000 | 13.385 | 13.385 | 13.385 | 13.385 | 13.385 | 13.385 | **20.077** | 26.769 |
| | | DICE - $wasserstein_{max}^{\gamma}$ | 0.000 | 13.385 | 13.385 | 13.385 | 13.385 | 13.385 | 13.385 | 13.385 | 10.038 |
| | | DICE - $inf_{mean}^{\gamma}$ | 0.000 | 13.385 | 13.385 | 13.385 | 13.385 | 13.385 | 13.385 | 13.385 | 13.385 |

Table 10: Rejection Quality of the top `SC-CE` combinations and Selective Classification baselines for the dataset `German` and `Adult`.

| Dataset | Black Box | Rejection Policy | Target Coverages | | | | | | | | |
| | | | 99% | 95% | 90% | 85% | 80% | 75% | 70% | 65% | 60% |
|---|---|---|---|---|---|---|---|---|---|---|---|
| Wisconsin | RF | PlugInRule | **0.000** | 13.700 | 5.871 | 6.850 | 4.215 | 3.806 | 3.425 | 3.114 | 2.322 |
| | | ILS - $cosine_{min}$ | **0.000** | 9.133 | 8.220 | 5.138 | 4.567 | 3.425 | 2.436 | 2.915 | 2.322 |
| | | $ILS_{latent}$ - $wasserstein^\gamma_{min}$ | **0.000** | 27.400 | 27.400 | **27.400** | **82.200** | **82.200** | **27.400** | 10.275 | 6.323 |
| | | $ILS_{latent}$ - $wasserstein^\gamma_{mean}$ | **0.000** | 0.000 | 27.400 | **27.400** | 54.800 | **82.200** | 20.550 | 11.743 | 4.982 |
| | | ILS - $sqeuclidean^\gamma_{max}$ | **0.000** | **82.200** | **41.100** | 27.400 | 16.440 | 13.700 | 13.700 | **13.700** | **13.700** |
| | | DICE - $cosine^\gamma_{mean}$ | **0.000** | 54.800 | 8.220 | 5.138 | 4.059 | 4.152 | 3.425 | 2.978 | 2.362 |
| | | $ILS_{latent}$ - $braycurtis^\gamma_{mean}$ | **0.000** | 13.700 | 10.275 | 8.220 | 6.323 | 5.871 | 5.871 | 5.138 | 4.326 |
| | mlp | PlugInRule | **0.000** | 0.000 | 5.792 | 3.564 | 2.574 | 2.648 | 3.390 | 3.159 | 2.317 |
| | | PlugInRuleAUC | **0.000** | 0.000 | 0.000 | 0.000 | 0.000 | 0.000 | 0.000 | 0.000 | 0.946 |
| | | LORE - $wasserstein_{min}$ | **0.000** | 92.667 | 18.533 | 10.296 | 5.148 | 2.896 | 3.658 | 3.310 | 2.623 |
| | | LORE - $wasserstein_{min}$ | **0.000** | 92.667 | 13.238 | 7.722 | 5.148 | 3.432 | 2.725 | 2.574 | **3.089** |
| | | LORE - $braycurtis_{min}$ | **0.000** | 46.333 | 11.583 | 4.212 | 4.633 | 3.159 | 3.022 | 2.673 | 2.482 |
| | | LORE - $sqeuclidean^\gamma_{min}$ | **0.000** | 23.167 | 7.722 | 7.722 | **11.583** | **6.318** | **4.344** | **3.564** | 2.482 |
| | | LORE - $l1^\gamma_{min}$ | **0.000** | 23.167 | 9.267 | **11.583** | 6.619 | 4.633 | 3.658 | 3.022 | 2.527 |
| | xgboost | PlugInRule | **0.000** | 30.889 | 15.444 | 4.877 | 2.317 | 2.780 | 2.574 | 2.317 | 2.014 |
| | | ILS - $cosine_{min}$ | **0.000** | 0.000 | 4.212 | 4.029 | 3.432 | 2.155 | 2.673 | 2.673 | 2.106 |
| | | $ILS_{latent}$ - $cosine^\gamma_{max}$ | **0.000** | 0.000 | **30.889** | **30.889** | 30.889 | 18.533 | 11.583 | 10.296 | 5.148 |
| | | $ILS_{latent}$ - $minkowski^\gamma_{max}$ | **0.000** | 23.167 | 18.533 | 11.583 | 9.267 | 8.424 | 8.424 | 7.316 | 3.658 |
| | | ILS - $inf^\gamma_{max}$ (D) | **0.000** | 46.333 | 23.167 | 15.444 | 13.238 | 7.128 | 4.029 | 2.206 | 1.495 |
| | | $ILS_{latent}$ - $l2^\gamma_{max}$ | **0.000** | 23.167 | 18.533 | 11.583 | 9.267 | 8.424 | 8.424 | 7.316 | 3.475 |
| | | $ILS_{latent}$ - $braycurtis^\gamma_{max}$ | **0.000** | 0.000 | 0.000 | 0.000 | 23.167 | **23.167** | **23.167** | **23.167** | 10.296 |
| | LGBM | PlugInRule | **0.000** | 51.750 | 5.175 | 4.500 | 3.234 | 2.654 | 2.587 | 2.112 | 2.226 |
| | | PlugInRuleAUC | **0.000** | 0.000 | 0.000 | 0.000 | 0.000 | 1.500 | 1.150 | 0.767 | 0.651 |
| | | $ILS_{latent}$ - $l1^\gamma_{max}$ | **0.000** | 51.750 | **51.750** | 25.875 | 17.250 | 14.786 | 12.938 | 5.750 | 2.797 |
| | | ILS - $wasserstein^\gamma_{max}$ | **0.000** | 0.000 | 0.000 | 0.000 | **34.500** | **69.000** | **69.000** | 34.500 | 20.700 |
| | | ILS - $sqeuclidean^\gamma_{max}$ | **0.000** | **69.000** | 34.500 | **51.750** | 17.250 | 7.962 | 5.175 | 2.875 | 1.643 |
| | | $ILS_{latent}$ - $sqeuclidean^\gamma_{mean}$ | **0.000** | 0.000 | 0.000 | 0.000 | 0.000 | 34.500 | 69.000 | 69.000 | **34.500** |
| | | ILS - $l2^\gamma_{max}$ (D) | **0.000** | 34.500 | **51.750** | 34.500 | 14.786 | 7.393 | 5.175 | 3.044 | 1.848 |
| Two-Moons | RF | PlugInRule | **0.000** | 10.905 | 6.415 | 4.050 | 4.241 | 3.793 | 3.862 | 3.289 | 2.596 |
| | | PlugInRuleAUC | **0.000** | 5.452 | 3.965 | 4.241 | 4.888 | 3.375 | 3.561 | 2.943 | 2.252 |
| | | DICE - $cosine^\gamma_{max}$ | **0.000** | 21.810 | **21.810** | **32.714** | 13.631 | 6.134 | 3.998 | 2.399 | 1.737 |
| | | $ILS_{latent}$ - $wasserstein^\gamma_{min}$ | **0.000** | 0.000 | 0.000 | 10.905 | 10.905 | **21.810** | 10.905 | 10.905 | **7.270** |
| | | $ILS_{latent}$ - $mae^\gamma_{min}$ | **0.000** | 0.000 | 0.000 | 10.905 | **21.810** | 21.810 | 7.270 | **10.905** | 7.270 |
| | | DICE - $sqeuclidean^\gamma_{mean}$ | **0.000** | **54.524** | 5.452 | 7.088 | 5.640 | 5.305 | 4.063 | 3.760 | 2.796 |
| | | DICE - $cosine^\gamma_{mean}$ | **0.000** | 0.000 | 0.000 | 21.810 | **21.810** | 4.089 | 3.635 | 2.019 | 1.410 |
| | mlp | PlugInRule | **0.000** | 10.364 | 9.144 | 8.008 | 6.662 | 4.731 | 4.030 | 3.455 | 2.651 |
| | | PlugInRuleAUC | **0.000** | 0.000 | 9.566 | 8.636 | 6.218 | **5.602** | 3.911 | 3.065 | 2.315 |
| | | DICE - $l1^\gamma_{max}$ | **0.000** | 72.545 | **20.727** | 8.008 | 5.330 | 4.190 | 3.636 | 3.508 | **2.780** |
| | | DICE - $l1_{max}$ | **0.000** | 93.273 | 19.432 | **12.584** | **8.008** | 5.042 | **4.230** | **3.636** | 2.651 |
| | | DICE - $wasserstein^\gamma_{max}$ | **0.000** | 72.545 | 17.273 | 8.390 | 5.330 | 3.969 | 3.574 | 3.201 | 2.714 |
| | | DICE - $wasserstein_{max}$ | **0.000** | 93.273 | 17.273 | 11.745 | 7.660 | 4.909 | 3.986 | 3.343 | 2.591 |
| | | DICE - $sqeuclidean^\gamma_{max}$ | **0.000** | 93.273 | 16.121 | 10.364 | 5.830 | 4.055 | 3.516 | 3.298 | **2.780** |
| | xgboost | PlugInRule | **0.000** | 12.889 | 6.444 | 4.603 | 3.580 | 2.578 | 2.181 | 2.539 | 1.862 |
| | | PlugInRuleAUC | **0.000** | 4.833 | 4.101 | 2.313 | 2.762 | 2.578 | 2.148 | 2.613 | 2.197 |
| | | DICE - $l2_{max}$ | **0.000** | **25.778** | **15.037** | **13.880** | 7.638 | 4.296 | **3.913** | 3.314 | **2.762** |
| | | DICE - $inf_{max}$ | **0.000** | **25.778** | 13.963 | 10.175 | **7.733** | **4.483** | 3.478 | 3.478 | 2.698 |
| | | DICE - $l2_{mean}$ | **0.000** | 19.333 | 12.889 | 10.311 | 6.652 | 4.124 | 3.381 | 3.314 | 2.729 |
| | | DICE - $sqeuclidean^\gamma_{max}$ | **0.000** | 19.333 | 12.889 | 8.056 | 5.728 | 4.044 | 3.652 | 3.362 | 2.607 |
| | | DICE - $sqeuclidean_{mean}$ | **0.000** | 17.185 | 11.815 | 12.889 | 6.652 | 4.124 | 3.437 | **3.412** | 2.729 |
| | LGBM | PlugInRule | **0.000** | 0.000 | 6.648 | 4.266 | 2.659 | 3.094 | 2.637 | 2.234 | 1.988 |
| | | PlugInRuleAUC | **0.000** | 1.125 | 1.009 | 1.625 | 1.590 | 2.437 | 2.216 | 2.041 | 2.089 |
| | | LORE - $cosine^\gamma_{max}$ | **0.000** | **29.250** | 12.188 | **12.188** | 6.094 | 4.301 | 3.179 | 2.812 | **2.925** |
| | | DICE - $l1_{max}$ | **0.000** | 7.312 | 9.402 | 8.603 | 6.435 | 5.432 | **4.875** | **4.062** | 2.819 |
| | | DICE - $wasserstein_{max}$ | **0.000** | 7.312 | 9.402 | 8.603 | 6.435 | **6.022** | 4.668 | 3.917 | 2.753 |
| | | DICE - $sqeuclidean^\gamma_{mean}$ | **0.000** | 9.750 | 8.775 | 5.388 | **6.500** | 5.850 | **4.875** | 3.849 | 2.708 |
| | | DICE - $sqeuclidean^\gamma_{max}$ | **0.000** | 8.775 | 8.357 | 6.187 | 5.941 | 4.477 | 3.718 | 3.274 | 2.721 |

Table 11: Rejection Quality of the top SC-CE combinations and Selective Classification baselines for the dataset Wisconsin and Two-Moons.

# I ADDITIONAL RESULTS FOR RQ3: SC–CE EXPLANATION IN CASE OF *accepted* INSTANCES

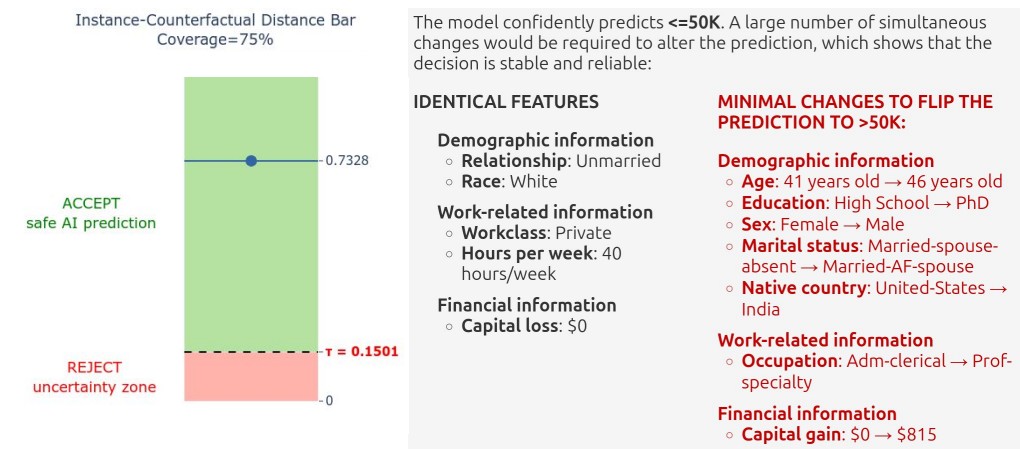

Figure 16: Example of explanation generated by SC–CE for an *accepted* test instance of the Adult dataset. The output includes a banner warning the user that the AI system can provide a prediction for the input since it is placed inside the model's safe zone, along with instructions on how to properly interpret the figures.

