# OpenReview forum: ""I know that I don’t know... and I explain why'' Interpretable abstention via counterfactual explanations"
_ICLR.cc/2026/Conference — Submitted to ICLR 2026_

### Official Review · Reviewer_D9An · 2025-10-26

**Soundness:** 2
**Presentation:** 2
**Contribution:** 2
**Rating:** 2
**Confidence:** 4

**Summary:**

This paper introduces a new framework for selective classification that leverages counterfactual explanations.
The main idea is to evaluate the uncertainty of a prediction by the minimum perturbation size required to flip it.
That is, the authors propose to use the minimum distance between an input and its counterfactual explanation as a proxy for uncertainty.
It enables us to explain why a classifier abstains from providing a prediction for the input.
The authors conducted experiments with several existing counterfactual explanation methods and different distance functions.
Experimental results suggest that the proposed method performed comparably to the baselines across the popular evaluation metrics in selective classification.

**Strengths:**

1. The authors tackle a well-motivated problem of the lack of explainability in selective classification. I think it is an intuitive and interesting idea to use the minimum distance between an input and its counterfactual explanation as a proxy for uncertainty.

**Weaknesses:**

1. While the idea of combining selective classification with counterfactual explanation is interesting, the proposed method appears to be a straightforward application of counterfactual explanation to the selective classification framework. Since it lacks technical novelty or theoretical analysis, I am not sure whether its technical contribution is sufficient for ICLR. For example, analyzing theoretical relationships between the conventional and the proposed methods (e.g., a condition under which they coincide) or identifying requirements that counterfactual explanations must satisfy to preserve the quality of selective classification would significantly strengthen the paper.
1. The presentation of this paper could be improved. Some paragraphs are overly long or include redundant descriptions, making the paper difficult to follow. In Section 4, the authors state, "From the table, we can see that generally using ILSlatent yields good and fairly consistent correlations with L2 distance, ..." but it is unclear how this conclusion was drawn from Table 2. Moreover, although various combinations of counterfactual explanation methods and distance functions were tested, Figure 2 is difficult to interpret: it is hard to tell which plot corresponds to which combination, and the selected combinations differ across panels. It would be clearer to present results for a few representative combinations in the main paper and place the complete results in the Appendix.

**Questions:**

1. As one of the distance functions, the authors employed the Wasserstein distance. While I acknowledge it is a distance between probability distributions, how do the authors use it to evaluate the distance between an input point and its counterfactual?

---

### Official Review · Reviewer_WJKH · 2025-10-27

**Soundness:** 3
**Presentation:** 3
**Contribution:** 3
**Rating:** 8
**Confidence:** 5

**Summary:**

The paper proposes an interpretable reject option based on the distance of the input to its closest counterfactual explanations.

**Strengths:**

- Relevant and well-motivated problem, which is grounded in literature
- Clear contribution with potentially high impact in practice
- Empirical evaluation is sound

**Weaknesses:**

- Repository looks a bit "unorganised". Please consider cleaning the requirements.txt and removing the To-Do list from the README. Is "OLD_src" really needed? Consider describing the structure of the repository, ...

Minor:
- Line 208 "Eq. equation 3" one "equation" must go xD
- References: For some references, a DOI is provided, while for others not. Please be consistent. I recommend always providing a DOI or some other permanent link to the cited publication

**Questions:**

None

---

### Official Review · Reviewer_o4Vf · 2025-10-29

**Soundness:** 3
**Presentation:** 3
**Contribution:** 2
**Rating:** 4
**Confidence:** 4

**Summary:**

The paper proposes Selective Classification via Counterfactual Explanations (SC‑CE), a plug‑in, model‑agnostic framework that uses the distance between an instance and its counterfactuals as a proxy for model confidence. The same counterfactuals are then displayed to the user when an instance is rejected, thereby providing a contrastive, actionable explanation of the abstention decision. The authors evaluate SC‑CE on four tabular datasets (Adult, German, Wisconsin, Two‑Moons) with four black‑box learners (LightGBM, MLP, Random Forest, XGBoost) and several counterfactual generators (DiCE, LoRE, ILS/ILS‑latent). They compare against two state‑of‑the‑art selective classifiers (PlugInRule, PlugInRuleAUC) using non‑rejected accuracy (NA), classification quality (CQ), and rejection quality (RQ). Empirically, SC‑CE achieves NA curves comparable to the baselines while offering interpretable explanations.

**Strengths:**

## Motivation and Relevance
Authors describe the well‑known trust-transparency trade‑off in human‑AI collaboration and identify abstention as a missing piece of interpretability. The introduction cites recent surveys (Mehrotra et al., 2024) that underscore the need for transparent uncertainty signals.

## Model‑agnostic design
SC‑CE only requires hard predictions from the black‑box learner. No gradient, probability score or internal architecture is needed—this is a strong selling point for real‑world deployment where models may be locked or proprietary.

## Unified use of counterfactuals
Counterfactuals are traditionally a post‑hoc explanation tool. SC‑CE repurposes them as both confidence and explanation, eliminating a second step. The authors argue that “a short counterfactual distance implies high uncertainty” (Section 3.2.1) and empirically show that the correlation between distance and probability is high (Table 2).

## User‑centric explanation format
The paper describes a multimodal output: bar plot of distance vs threshold and the counterfactual itself. This aligns with human‑centric XAI literature (Miller 2019) that stresses contrastive, actionable explanations.

**Weaknesses:**

## Limited novelty
The core idea of using counterfactual distance as a confidence proxy is not new. Prior work (Artelt et al., 2023; Singla et al., 2023) has already suggested that “easy” counterfactuals indicate uncertainty. The manuscript does not provide a theoretical analysis or empirical evidence that SC‑CE’s distance measure is superior to existing uncertainty estimates (e.g., probability scores, entropy).

## Narrow baseline set
Only two plug‑in baselines (PlugInRule, PlugInRuleAUC) are evaluated. Modern selective classification benchmarks include conformal prediction (Hallberg Szabadváry et al., 2025), learned rejectors (Pugnana & Ruggieri, 2023), and deep‑ensemble uncertainty. Without these, it is unclear whether SC‑CE offers a tangible advantage over state‑of‑the‑art methods.

## No statistical edge
The Friedman test reports no significant difference between SC‑CE and the best baseline in NA (p > 0.05). Yet the paper claims “SC‑CE matches or surpasses” without clarifying that the advantage is marginal. A more balanced discussion would be appropriate.

## Missing interpretability validation
The central claim is that SC‑CE provides useful explanations. The paper only shows screenshots (Figure 16). No user study, no quantitative counterfactual metric (e.g., plausibility, proximity, sparsity), and no comparison to other explanation methods.

## Computational overhead unreported
Counterfactual generation (DiCE, LoRE, ILS) can be expensive, especially in high‑dimensional settings. The manuscript does not report runtimes or memory consumption, nor compare them to the lightweight baselines that require only a threshold on probabilities.

## Hyperparameter sensitivity ignored
SC‑CE depends on several design choices: number of counterfactuals per instance, choice of distance metric, aggregation function (min/mean/max), and threshold calibration (empirical percentile vs Gamma). No ablation study is presented to show how sensitive NA and RQ are to these choices.

## Limited data scope
All experiments are on small tabular datasets (≤ 50 k instances, ≤ 30 features). No experiments on high‑dimensional tabular data (≥ 100 features) or on images/text where counterfactual generation is more challenging. Thus generality of the claims is uncertain.

**Questions:**

1. Can you provide a theoretical justification that counterfactual distance is a better uncertainty estimate than existing probability‑based or entropy measures?

2. The literature distinguishes several possibilities for counterfactuals: (i) those that merely approximate the decision boundary (Wachter 2017), (ii) plausible counterfactuals that lie on the data manifold and might be far from the decision boundary (Wielopolski 2024), (iii) highly diverse counterfactuals spanning the manifold (DiCE). How does SC‑CE’s performance (NA, CQ, RQ) vary under these different counterfactual regimes? Please discuss the implications of each regime for both uncertainty estimation and explanation quality. Furthermore, I think it would be interesting to see how SC-CE behaves when counterfactuals are substituted with simply k-nn from opposite class.

3. Have you considered integrating global counterfactual explanations, methods that generate a single change‑vector direction for the whole model (e.g., GLOBE‑CE)? If so, how would such a global approach affect the design of SC‑CE (e.g., threshold calibration, explanation richness) and its interpretability?

3. Have you performed any user study or applied quantitative XAI metrics (faithfulness, stability) to assess the usefulness of the counterfactual explanations? If not, could you plan a small experiment with domain experts (e.g., in finance or healthcare) to validate the claims?

4. What is the average time to generate counterfactuals per instance for each generator? How does this compare to the cost of a simple probability‑threshold rejector? Could you include a runtime table in the appendix or supplementary material?

---

### Official Review · Reviewer_Qxv9 · 2025-11-03

**Soundness:** 2
**Presentation:** 2
**Contribution:** 2
**Rating:** 2
**Confidence:** 4

**Summary:**

The paper proposes to use counterfactual explanation techniques to explain predictions that a selective classifier abstains on. The authors propose to use distance to nearest counterfactual point as a measure of uncertainty, which the selective classifier uses to decide on whether to abstain or not.

**Strengths:**

- Paper is generally well written and clear. Self-contained
	- e.g., appreciate explanation of metrics in Section 4
- Some of the experiments answer questions one might have when reading the paper
	- e.g., experiment answering RQ1

**Weaknesses:**

**Weak problem statement**

The authors suggest that not having explanations for abstention decisions in selective classifiers is a problem:

> Given that abstention mechanisms directly affect subsequent human workflows, the provision of interpretable rejections is essential: individuals require not only awareness of a model’s uncertainty but also understanding of the underlying reasons for abstention, enabling them to determine whether to accept, contest, or act on the deferred case, without compromising trust in the AI system (lines 45-49).

Why is it essential that we have interpretable rejections? What gains does it provide? How exactly do explanations help decision-makers determine how to act on a deferred case? How does the lack of explanations compromise trust? There is a lot to unpack here.

I believe the authors will be able to demonstrate this through concrete use cases.

**Analysis and discussion of results**

There is a hint of this in Figure 3 and the paragraph answering RQ3 (line 422). However, the authors just show that it is "interpretable"—without any discussion on what this might mean for the end user.

Also, the discussion of results in Table 2 (line 341) should reference specific numbers in the table instead of saying:
> yields good and fairly consistent correlations (line 346)

In fact, this discussion on the extent to which the distance from the nearest counterfactual maps to uncertainty (predictive probability) is arguably the most important experimental result in the paper; the validity of SC-CE hinges on this. The paper lacks thorough discussion here. The authors just state that:

> Naturally, this heavily depends on the shape of the decision boundary of the classifier, as highlighted by the differences between the different models for the same dataset (lines 348-349).

-  What does this suggest? (e.g., what does this say about the validity of the procedure?)
-  What procedure should one follow in selecting distance measures?
-  At what point can we say it is appropriate to use distance as a proxy metric for uncertainty?

One could imagine just using traditional rejection and explaining those predictions. How might this compare with SC-CE? The authors need to present a way to compare these predictions and explanations, which requires thinking about downstream tasks for explaining rejections.

The empirical results in Figure 2 are difficult to parse. I would suggest the authors reduce the number of examples they show (put the rest in the appendix) and select salient methods that highlight what they want to discuss.

**Questions:**

- I found it interesting that SC-CE framework allows different distance metrics when aggregating counterfactual points (line 202-212). When might one use a different distance metric when computing $c_d(x)$?

- > Specifically, for each rejected instance, SC-CE provides a multi-modal explanation that combines textual and visual elements

Why is this a salient point?

---

### Meta-Review · Area_Chair_kEYX · 2026-01-06

**Summary:**

The paper proposes SC-CE, a selective classifier that uses counterfactual distance both as an uncertainty proxy and as an explanation for abstention. One reviewer recommends acceptance as the problem is important and the method is intuitive and practically appealing.  However, others raise substantial concerns regarding:

- Lack of convincing justification that counterfactual distance is a reliable uncertainty estimate
- Absence of evidence that the provided explanations are actually useful to users
- Limited novelty and benchmarking relative to modern uncertainty-aware methods

**Reviewer Concerns:**

The authors didn't provide rebuttal.

**Reviewer Scores:**

The authors didn't provide rebuttal. Thus, I don't expect the reviewers who gave negative ratings would have changed their score significantly. I feel that the reviewer who gave 8 might lower their rating.

---

### Decision · Program_Chairs · 2026-01-26

Reject